



**ERA-5 and ERA-Interim driven ISBA land surface model simulations : Which one performs better?**

Clement Albergel[1], Emanuel Dutra [2], Simon Munier[1], Jean-Christophe Calvet[1], Joaquin Munoz-Sabater[3], Patricia de Rosnay[3], Gianpaolo Balsamo[3]

[1]CNRM UMR 3589, Météo-France/CNRS, Toulouse, France

[2]Instituto Dom Luiz, IDL, Faculty of Sciences, University of Lisbon, Portugal

[3]ECMWF, Reading, UK

**Corresponding author :** Clement Albergel, clement.albergel@meteo.fr

**Abstract –** The European Centre for Medium Range Weather Forecast (ECMWF) recently released a first 7-year segment of its latest atmospheric reanalysis: ERA-5 over 2010-2016. ERA-5 important changes relative to ERA-Interim former atmospheric reanalysis include a higher spatial and temporal resolution as well as a more recent model and data assimilation system. ERA-5 is foreseen to replace ERA-Interim reanalysis and one of the main goals of this study is to assess whether ERA-

5 can enhance the simulation performances with respect to ERA-Interim when it is used to force a Land-Surface-Model (LSM). To that end, both ERA-5 and ERA-Interim are used to force the ISBA (Interactions between Soil, Biosphere, and Atmosphere) LSM fully coupled with the Total Runoff Integrating Pathways (TRIP) scheme adapted for the CNRM (Centre National de Recherches Météorologiques) continental hydrological system within the SURFEX (SURFace Externalisée)

modelling platform of Météo-France. Simulations cover the 2010-2016 period at half a degree spatial resolution.

ERA-5 impact on the ISBA LSM with respect to ERA-Interim is assessed over a data-rich area: North America. A comprehensive evaluation of ERA-5 impact is conducted using remote sensing and in-situ observations covering a substantial part of the land surface storage and fluxes. The

remote sensing observations include: (i) satellite-driven model estimates of land evapotranspiration , (ii) upscaled ground-based observations of gross primary productivity, (iii) satellite derived estimates of surface soil moisture as well as (iv) satellite derived estimates of Leaf Area Index. The in-situ observations cover (i) soil moisture, (ii) turbulent heat fluxes and Net Ecosystem Exchange (NEE), (iii) river discharges and (iv) snow depth. ERA-5 leads to a consistent

improvement over ERA-Interim as verified with the use of these 8 independent observations of different land status and of the model simulations forced by ERA-5 when compared with ERA-Interim.. This is particularly evident for the land surface variables linked to the terrestrial hydrological cycle while variables linked to vegetation are less impacted. Results also indicate that while precipitation provides, to a large extend, improvements in surface fields (e.g. large



improvement in the representation of river discharge and snow depth), the other atmospheric variables play an important role, contributing to the overall improvements. These results highlight the importance of enhanced meteorological forcing quality provided by the new ERA-5 reanalysis, which will pave the way for a new generation of land-surface developments and applications.

1. **Introduction**

Observing and simulating the response of land biophysical variables to extreme events is a major scientific challenge in relation to the adaptation to climate change. To that end, Land Surface Models (LSMs) constrained by high quality gridded atmospheric variables and coupled with river routing models are essentials (Schellekens et al., 2017, Dirmeyer et al., 2006). Such LSMs should

represent land surface biogeophysical variables like surface and root zone soil moisture (SSM and RZSM, respectively), biomass and Leaf Area Index (LAI) in a way that is fully consistent with the representation of surface and energy flux as well as river discharge simulations. Land surface simulations, such as those from the Global Soil Wetness Project (GSWP, Dirmeyer et al., 2002, 2006; Dirmeyer, 2011), combined with seasonal forecasting systems have been of paramount

importance in triggering progresses in land-related predictability as documented in the Global Land–Atmosphere Coupling Experiments (GLACE, Koster et al., 2009a, 2011). The land surface state estimates used in those studies were generally obtained with offline (or stand-alone) model simulations, forced by 3-hourly meteorological fields from atmospheric reanalysis. In the past decade several improved global atmospheric reanalysis of the satellite era (1979-onwards) have

been produced that enable new applications of offline land surface simulations. Amongst them are NASA's Modern Era Retrospective-analysis for Research and Applications (MERRA; Rienecker et al., 2011) as well as ECMWF's (European Centre for Medium-Range Weather Forecasts) Interim reanalysis (ERA-Interim; Dee et al., 2011). Their offline use in LSMs led to global land surface variables (LSVs) reanalysis datasets that can support e.g. water resources analysis (Schellekens et

al., 2017), like MERRA-Land (Reichle, 2011) and ERA-Interim/Land (Balsamo et al., 2015).

The quality of those offline land surface simulations relies on the accuracy of the forcing and of the realism of the land surface model itself (Balsamo et al., 2015). ECMWF recently released the first 7-year segment of its latest atmospheric reanalysis: ERA-5 over 2010-2016. ERA-5 important changes relative to ERA-Interim former atmospheric reanalysis include a higher spatial and

temporal resolution as well as a better global balance of precipitation and evaporation. It will eventually replace ERA-Interim reanalysis. Assessing ERA-5 ability to force a LSM with respect to ERA-Interim is therefore highly relevant. To that end, ERA-5, ERA-Interim as well as a combination of both (ERA-5 with precipitation of ERA-Interim) are used to constrain the $CO_2$-



responsive version of the Interactions between Soil, Biosphere, and Atmosphere (ISBA, Noilhan
and Mahfouf, 1996; Calvet et al., 1998, 2004 ; Gibelin et al., 2006) LSM fully coupled with the
CNRM (Centre National de Recherches Météorologiques) version of the Total Runoff Integrating
Pathways (TRIP, Oki et al., 1998) continental hydrological system (CTRIP hereafter, Decharme et
al., 2010) within the SURFEX (SURFace Externalisée, Masson et al., 2013) modeling system of
Météo-France. ISBA models leaf-scale physiological processes and plant growth, while transfer of
water and heat through the soil rely on a multilayer diffusion scheme.

In this study SURFEX is applied over a data-rich area: North America (longitudes from 130.0ºW to
60.0º E, latitudes from 20.0º to 55.0º N) for the period 2010-2016. ERA-5 added values with respect
to ERA-Interim is assessed by providing verification and diagnostics comparing ISBA land surface
variables outputs when forced by either ERA-5, ERA-Interim, ERA-5 with ERA-Interim
precipitations to several in-situ measurement data sets as well as satellite-derived estimates of Earth
Observations. Namely, In-situ measurements of (i) soil moisture from the USCRN network (US
Climate Reference Network, Bell et al., 2013) spanning all over the United States of America and
(ii) of turbulent heat fluxes and Net Ecosystem Exchange (NEE) from FLUXNET-2015
(http://fluxnet.fluxdata.org/data/fluxnet2015-dataset/) are used in the evaluation, together with (iii)
river discharges from the United States Geophysical Survey (USGS, https://waterwatch.usgs.gov/)
and (iv) snow depth measurements from the Global Historical Climatology Network (GHCN,
Menne et al., 2012a, 2012b). Are also used: (i) satellite-driven model estimates of land
evapotranspiration from the Global Land Evaporation Amsterdam Model (GLEAM, Martens et al.,
2017) project, (ii) upscaled ground-based observations of gross primary productivity from the
FLUXCOM project (Jung et al., 2017), (iii) satellite derived estimates of SSM from the Climate
Change Initiative (CCI) program of the European Space Agency (ESA-CCI SSMv4 Dorigo et al.,
2015, 2017) as well as (iv) satellite derived estimates of LAI from the Copernicus Global Land
Service program (CGLS, http://land.copernicus.eu/global/).

Section 2 presents the two atmospheric reanalyses data sets, ERA-Interim and ERA-5, the SURFEX
model configuration as well as the evaluation strategy with the observational data sets. Section 3
provides a set of statistical diagnostics to assess and evaluate ERA-5 impact in ISBA with respect to
ERA-Interim. Finally, section 4 provides perspectives and future research directions.

## 2. **Methodology**

### 2.1.   *ERA-Interim and ERA-5 reanalysis*

ERA-Interim is a global atmospheric reanalysis produced by ECMWF (Dee et al. 2011). It uses
Integrated   Forecast   System   (IFS)   version   31r1   (more   information   at



https://www.ecmwf.int/en/forecasts/documentation-and-support/changes-ecmwf-model/ifs-documentation) with a spatial resolution of about 80 km (T255) and with analyses available for

0000, 0600, 1200, and 1800 UTC. It covers the period from 1 January 1979 onward and continues to be extended forward in near-real time (with a delay of approximately 1 month). Reanalyses merge observations and model forecasts in data assimilation methods to provide an accurate and reliable description of the climate over the last few decades. Berrisford et al. (2009) provide a detailed description of the ERA-Interim product archive. ERA-5 (Hersbach, 2016) is the latest and

fifth generation of European reanalyses produced by the ECMWF and a key element of the EC-funded Copernicus Climate Change Service (C3S). It is expected that ERA-5 will replace the production of the current ERA-Interim reanalysis (Dee et al., 2011) before the end of 2018, from 1979 to close to Near Real Time (NRT) period, i.e., in ERA-5 regular routine updates will be conducted to keep close to NRT. In a second phase, an extension back to 1950 is also expected.

ERA-5 adds different characteristics to ERA-Interim reanalysis, which makes it richer in term of climate information.

ERA-5 uses one of the most recent versions of the Earth system model and data assimilation methods applied at ECMWF, which makes it able to use modern parameterizations of Earth processes compared to older versions used in ERA-Interim. For instance, developments done at

ECMWF allows ERA-5 to apply a variational bias scheme not only to satellite observations, but also to ozone, aircraft and surface pressure data. ERA-5 benefits also of reprocessed data sets that were not ready yet during the production of ERA-Interim. Two other important features of ERA-5 are the improved temporal and spatial resolution, from 6-hourly in ERA-Interim to hourly analysis in ERA-5, and from 79 km in the horizontal dimension and 60 levels in the vertical, to 31 km and

137 levels in ERA-5. Finally, ERA-5 also provides an estimate of uncertainty through the use of a 10-member Ensemble of Data Assimilations (EDA) at a coarser resolution (63 km horizontal resolution) and 3-hourly frequency.

### 2.2.    *SURFEX modeling system*

#### 2.2.1.    *The ISBA Land-Surface-Model*

In this study the $CO_2$-responsive version ISBA LSM is included in the open-access SURFEX modelling platform of Météo-France (Masson et al., 2013). The most recent version of SURFEX (version 8.1) is used in this study with the "NIT" biomass option for ISBA. The latter simulates the diurnal cycle of water and carbon fluxes, plant growth and key vegetation variables like LAI and above ground biomass on a daily basis. It can be coupled to the CTRIP river routing model in order

to simulate the streamflow. In this version of ISBA, a single-source energy budget of a soil–vegetation composite is computed. Also, the ISBA parameters are defined for 12 generic land





surface patches, which include nine plant functional types (needle leaf trees, evergreen broadleaf trees, deciduous broadleaf trees, C3 crops, C4 crops, C4 irrigated crops, herbaceous, tropical herbaceous, and wetlands), bare soil, rocks, and permanent snow and ice surfaces. A more

comprehensive model description can be found in Masson et al. (2013).

ISBA accounts for the atmospheric $CO_2$ concentration on stomatal aperture (Calvet et al., 1998, 2004; Gibelin et al., 2006). Also, photosynthesis and its coupling with stomatal conductance at a leaf level are accounted for. The vegetation net assimilation of $CO_2$ is estimated and used as an input to a simple vegetation growth sub-model able to predict LAI: photosynthesis drives the

dynamic evolution of the vegetation biomass and LAI variables in response to atmospheric and climate conditions. During the growing phase, enhanced photosynthesis corresponds to a $CO_2$ uptake, which leads to vegetation growth. In contrast, lack of photosynthesis leads to higher mortality rates. The gross primary production (GPP) is defined as the carbon uptake while the ecosystem respiration (RECO) is the release of $CO_2$, the difference between these two quantities

being the net ecosystem $CO_2$ exchange (NEE). Evaporation due to (i) plant transpiration, (ii) liquid water intercepted by leaves, (iii) liquid water contained in top soil layers, and (iv) the sublimation of the snow and soil ice are combined to represent the total evaporative flux.

ISBA 12-layers explicit snow scheme (Boon et Etchevers, 2001, Decharme et al., 2016) as well as its multi-layer soil diffusion scheme (ISBA-Dif) are used. The later is based on the mixed form of

the Richards equation (Richards, 1931) and explicitly solves the one-dimensional Fourier law. It also incorporates soil freezing processes developed by Boone et al. (2000) and Decharme et al. (2013). The total soil profile is vertically discretized; both the temperature and moisture of each soil layer are computed accordingly to their textural and hydrological characteristics. The Brookes and Corey model (Brooks and Corey, 1966) determines the closed-form equations between the soil

moisture and the soil hydrodynamic parameters, including the hydraulic conductivity and the soil matrix potential (Decharme et al., 2013). The defaults discretization with 14 layers over 12 m depth is used. The lower boundary of each layer being: 0.01, 0.04, 0.1, 0.2, 0.4, 0.6, 0.8, 1, 1.5,2, 3, 5, 8 and 12 m deep (see Fig. 1 of Decharme et al., 2011). Amounts of clay, sand and organic carbon in the soil determine the by thermal and hydrodynamic soil properties (Decharme et al., 2016). They

are taken from the Harmonised World Soil Database (HWSD, Wieder et al., 2014). As for hydrology, the infiltration, surface evaporation and total runoff are accounted for in the soil water balance. The infiltration rate defines the discrepancy between the surface runoff and the throughfall rate. The later being defined as the sum of rainfall not intercepted by the canopy, dripping from the canopy (i.e., interception reservoir) as well as snow melt water. The soil evaporation affects only the

superficial layer (top 1 cm) and is proportional to its relative humidity. Transpiration water from the



root zone (the region where the roots are asymptotically distributed) follows the equations in Jackson et al. (1996). Canal et al. (2014) provide more information on the root density profile.

Both the surface runoff (the lateral subsurface flow in the topsoil) and a free drainage condition at the bottom soil layer contribute to ISBA total runoff.

The Dunne runoff (i.e. when no further soil moisture storage is available) and lateral subsurface flow from a sub-grid distribution of the topography are computed using a basic TOPMODEL approach. The Horton runoff (i.e. when rainfall has exceeded infiltration capacity) is estimated from the maximum soil infiltration capacity and a sub-grid exponential distribution of the rainfall intensity.

2.2.2.    *The CTRIP hydrological system*

CTRIP is driven by three prognostic equations corresponding to (i) the groundwater, (ii) the surface stream water and (ii) the seasonal floodplains. Streamflow velocity is computed using the Manning formula as described in Decharme et al. (2010). When the river water level overtops the river-bank, it fills up the floodplain reservoir which empties when the water level drops below this threshold

(Decharme et al., 2012). Occurrence of flooding impacts the ISBA soil hydrology through infiltration and it influences the overlying atmosphere via free surface water evaporation and precipitation interception, also. The groundwater scheme is based on the two-dimensional groundwater flow equation for the piezometric head (Vergnes and Decharme,2012). Its coupling with ISBA enables accounting for the presence of a water table under the soil moisture column. It

allows upward capillary fluxes into the soil (Vergnes et al., 2014). CTRIP is coupled to ISBA through OASIS-MCT (Voldoire et al., 2017). Once a day, ISBA provides CTRIP with updates on runoff, drainage, groundwater and floodplain recharges, and CTRIP feedbacks to ISBA the water table depth or rise, floodplain fraction, and flood potential infiltration. The current CTRIP version consists of a global streamflow network at 0.5° spatial resolution.

2.3.    *Evaluation strategy and data sets*

Three experiments are considered for the evaluation; (i) SURFEX forced by ERA-Interim, all atmospheric variables interpolated at 0.5°x0.5° spatial resolution (referred as ei_S hereafter, the benchmark experiment), (ii) SURFEX forced by ERA-5 all atmospheric variables interpolated at 0.5°x0.5° spatial resolution except precipitation (rain and snow interpolated to hourly time steps

assuming a constant flux) that comes from ERA-Interim (referred as e5ei_S hereafter). (iii) SURFEX forced by ERA-5, all atmospheric variables interpolated at 0.5°x0.5° spatial resolution (referred as e5_S hereafter). For all three experiments, the first year (2010) was spun up 20 times to allow the model to reach equilibrium. Comparing e5_S to ei_S provides the overall improvements



from ERAI to ERA-5. The idealized e5ei_S simulation was carried out to assess the role of
precipitation changes from ERA-Interim to ERA-5.

This study makes use of several in-situ measurement data sets as well as satellite-derived estimates
of Earth Observations that are described in the next two sections. The different performance metrics
used for the evaluation are described, also. Their choice is of crucial interest; it is governed by the
nature of the variable itself and is influenced by the purpose of the investigation and its sensitivity
to the considered variables (Stanski et al. 1989). No single metric or statistic can capture all the
attributes of environmental variables. Some are robust in respect to some attributes while insensitive
to others (Entekhabi et al. 2010). While performance metrics like the correlation coefficient, mean
absolute error, unbiased root mean squared differences, root mean squared differences, efficiency
score (depending on the considered variable) are first applied to the three simulations
independently, metrics like the normalized information contribution (e.g. Kumar et al., 2009) are
then used to quantify improvement or degradation from a data set to another. Table I summarises the
different dataset used for the evaluation as well as the performance metrics used.

### 2.3.1.    *In situ measurement of soil moisture, river discharges, snow depth and fluxes*

USCRN is a network of climate-monitoring stations maintained and operated by the National
Oceanic and Atmospheric Administration (NOAA). It aims at providing climate-science-quality
measurements of air temperature and precipitation. To increase the network's capability of
monitoring soil processes and drought, soil observations were added to USCRN instrumentation. In
2011, the USCRN team completed at each USCRN station in the conterminous United States the
installation of triplicate-configuration soil moisture and soil temperature probes at five standards
depths (5, 10, 20, 50, and 100 cm) as prescribed by the World Meteorological Organization. 111
stations present data between 2009 and 2016. Stations provide data at an hourly time step. Similar
to prior study, datasets potentially affected by frozen conditions were masked out using an observed
temperature threshold of 4ºC (e.g. Albergel et al., 2013a). The second layer of soil of ISBA between
1 and 4 cm depth (the diffusion scheme is used in this study) is compared to in situ measurements at
5 cm depth at a three hourly time step (model output) between April and September in order to
avoid as much as possible frozen conditions. The ability of ei_S, e5ei_S and ei_S to reproduce
surface soil moisture variability is first assessed using the correlation coefficient (R) and unbiased
Root Mean Square Differences (ubRMSD). Climatology differences between model and in-situ
observations make a direct comparison difficult (Koster et al., 2009b). Soil moisture time series
usually show a strong seasonal pattern possibly artificially increasing the perceived agreement
between modeled and observed data sets. To avoid seasonal effects, time series of anomalies from a
moving monthly averaged are also computed. At each grid and observation points, the difference to



the mean is calculated using a sliding window of five week and the difference is scaled by the standard deviation as in Albergel et al., (2013b). Anomaly time series reflect the time-integrated

impact of antecedent meteorological forcing. The latter is mainly reflected in the upper layer of soil. The correlation coefficient is also computed for anomaly time-series ($R_{ano}$). The p-value (a measure of the correlation significance) is also calculated indicating the significance of the test (as in Albergel et al. 2010), and only cases where the p-value is below 0.05 (i.e., the correlation is not a coincidence) are retained. Stations with non-significant R values can be considered suspect and are

excluded from the computation of the network average metrics. This process may remove some reliable stations too (e.g., in areas where the model might not realistically represent soil moisture). Over 2010-2016 river discharge from ei_S, e5ei_S and e5_S are compared to daily streamflow data from the U.S. Geological Survey (USGS; http://nwis.waterdata.usgs.gov/nwis). Data are chosen for sub-basins with large drainage areas (10,000km$^2$ or greater) and with a long observation time series

(4 years or more). Smaller basins are excluded due to the low resolution of CTRIP (0.5°). It is common to express observed and simulated river discharge (Q) data in m$^3$s$^{-1}$. Given that the observed drainage areas may differ slightly from the simulated ones, specific discharge in mm.d$^{-1}$ (the ratio of Q to the drainage area) is used in this study, similarly to Albergel et al., 2017. Stations with drainage areas differing by more than 20% from the simulated ones are also discarded. Impact

on Q is evaluated using the efficiency score (NSE) (Nash and Sutcliff, 1970). NSE evaluates the model ability to represent the monthly discharge dynamics and is given by:

$$NSE = 1 - \frac{\sum_{t=1}^{T} \left( Q_s^t - Q_o^t \right)^2}{\sum_{t=1}^{T} \left( Q_o^t - \overline{Q_o^t} \right)^2} \qquad (1)$$

where   $Q_s^t$   is the simulated river discharge (by either ei_S, e5ei_S or e5) at time t and   $Q_o^t$   is observed river discharge at time t, T is the total number of days and   $\overline{Q_o^t}$   is the average observed

discharge. NSE can vary between −∞ and 1. A value of 1 corresponds to identical model predictions and observed data. A value of 0 implies that the model predictions have the same accuracy as the the mean of the observed data. Negative values indicate that the observed mean is a more accurate predictor than the model simulation. Only stations with an NSE greater than -1 for the benchmark experiment, ei_S, are considered, leading to 172 stations over the considered domain. A normalized

information contribution (NIC as in Kumar et al. ; 2009) measure is then computed to quantify the improvement or degradation due to the specific atmospheric reanalysis used to force ISBA. The $NIC_{NSE}$   values are computed for both e5_S and e5ei_S with respect to ei_S as:





$$NIC_{NSE(e5;5ei)} = \frac{NSE_{(e5;e5ei)} - NSE_{(ei)}}{1 - NSE_{(ei)}} \qquad (2)$$

The $NIC_{NSE}$ metric provides a normalized measure of the improvement through the use of either

$NSE_{e5ei}$ or $NSE_{e5}$ as a fraction of the maximum possible skill improvement (1-$NSE_{ei}$). Positive and

negative $NIC_{NSE}$ values indicate improvements and degradations in either e5_S or e5ei_S relative to

ei_S river discharges estimates, respectively. The ratio of simulated and observed river discharges is

computed also $\left(Q_s^t/Q_o^t\right)$, the closer to one it is, the better the simulated river discharges are.

The Global Historical Climatology Network (GHCN) Daily dataset, developed to meet the needs of

climate analysis and monitoring studies that require data at a daily time resolution contains records

from over 75000 stations in 179 countries and territories (Menne et al., 2012a, b). Numerous daily

variables are provided, including maximum and minimum temperature, total daily precipitation,

snowfall, and snow depth. In this study, over North America, stations with daily snow depth data

from 2010-2016, with less than 10% missing and at least 15 days of snow presences per year on

average (to avoid using stations always reporting zero snow depth) are used, it is about 2000

stations. The ability of ei_S, e5ei_S and e5_S to reproduce snow depth and its variability is assessed

using the correlation coefficient (R), unbiased Root Mean Square Difference (ubRMSD) and mean

absolute error (MAE). In order to provide an easier measure of the added value of e5_S and e5ei_S,

statistics are also normalized with respect to ei_S, NIC is used for R:

$$NIC_{R(e5;e5ei)} = \frac{R_{(e5;e5ei)} - R_{(ei)}}{1 - R_{(ei)}} \qquad (3)$$

Normalized MAE, $N_{MAE}$, and ubRMSD, $N_{ubRMSD}$, are respectively computed as follow:

$$N_{MAE(e5;e5ei)} = 100 * \frac{MAE_{(e5ei;ei)} - MAE_{(ei)}}{MAE_{(ei)}} \qquad (4)$$

$$N_{ubRMSD(e5;e5ei)} = 100 * \frac{ubRMSD_{(e5ei;ei)} - ubRMSD_{(ei)}}{ubRMSD_{(ei)}} \qquad (5)$$

Daily observations of sensible and latent heat fluxes from the FLUXNET-2015 dataset with at least

2-yr of data are used over 2010-2016 to evaluate e5_S, e5ei_S and ei_S ability to reproduce flux

variability. The FLUXNET-2015 dataset includes data collected at sites from multiple regional flux

networks as well as several improvements to the data quality control protocols and the data

processing pipeline (http://fluxnet.fluxdata.org/data/fluxnet2015-dataset/). 37 stations are retained

for the evaluations, two metrics are considered: R and ubRMSD. Eq.(3) is also applied as well as

Eq.(5) on ubRMSD.



Performance metrics are applied to each individual station of each network; thereafter, network-average metrics are computed by averaging the statistics from the individual stations within each network.

2.3.2.     *satellite derived estimates of surface soil moisture, leaf area index, land evapotranspiration and gross primary production*

In response to the GCOS endorsement of soil moisture as an essential climate variable, the European Space Agency Water Cycle Multimission Observation Strategy (WACMOS) project and Climate Change Initiative (CCI; http://www.esa-soilmoisture-cci.org) have supported the generation

of a surface soil moisture product based on multiple microwave sources (ESA-CCI SSM hereafter). The first version of the combined product was released in June 2012 by the Vienna University of Technology (Liu et al. 2011, 2012; Wagner et al., 2012). Several authors (e.g. Albergel et al., 2013a, b; Dorigo et al., 2015, 2017) have highlighted the quality and stability over time of the product. Despite some limitations, this data set has already shown potential in assessing model performance

(Szczypta et al., 2014; van der Schrier et al., 2013). In this study the ESA CCI SM-combined latest version of the product (v4) which merges SM observations from seven microwave radiometers (SMMR, SSM/I, TMI, ASMR-E, WindSat, AMSR2, SMOS) and four scatterometers (ERS-1 and 2 AMI and MetOp-A and B ASCAT) into a combined data set covering the period November 1978 to December 2016. Data are in volumetric ($m^3m^{-3}$) units and quality flags (snow coverage or

temperature below 0° and dense vegetation) are provided. For a more comprehensive overview of the product see Dorigo et al. (2015, 2017). As topographic relief is known to negatively affect remote sensing estimates of soil moisture (Mätzler and Standley, 2000), the time series for pixels whose average altitude exceeded 1500 m above sea level were discarded. Data on pixels with urban land cover fractions larger than 15% were also discarded, to limit the effects of artificial surfaces.

The altitude and urban area thresholds were set according to Draper et al. (2011) and Barbu et al. (2014), who processed satellite-based SSM retrievals for data assimilation exercises with the ISBA LSM. As for in situ measurements of soil moisture, correlation is applied on both the volumetric and anomaly time series.

The GEOV1 LAI used in this study is produced by the European Copernicus Global Land Service

project (http://land.copernicus.eu/global/) as evaluated in Boussetta et al. (2015). The LAI observations are retrieved from the SPOT-VGT and then PROBA-V (from 1999 to present) satellite data according to the methodology proposed by Baret et al. (2013). As in Barbu et al. (2014), the 1 km spatial resolution observations are interpolated by an arithmetic average to the 0.5° model grid points, if at least 50 % of the observation grid points are observed (i.e half the maximum amount).

LAI observations have a temporal frequency of 10 days at best (in presence of clouds no



observations are available). Correlation and root mean squared differences are used to assess ei_S, e5ei_S and e5_S ability to reproduce LAI variability. Eq.(3) (NIC$_R$) as well as Normalized RMSD :

$$N_{RMSD(e5;e5ei)} = 100 * \frac{RMSD_{(e5ei;ei)} - RMSD_{(ei)}}{RMSD_{(ei)}} \qquad (7)$$

are used, also.

The GLEAM product uses a set of algorithms to estimate both terrestrial evaporation and root-zone SM based on satellite data (Miralles et al., 2011). It is a useful validation tool to assess model performance given that such quantities are difficult to measure directly on large scales. Potential evaporation rates are constrained by satellite-derived SM data while the global evaporation model in GLEAM is mainly driven by various microwave remote-sensing observations. It is now a well-

established dataset that has been widely used to study land–atmosphere feedbacks (e.g. Miralles et al., 2014b; Guillod et al., 2015) as well as trends and spatial variability in the hydrological cycle (e.g. Jasechko et al., 2013; Greve et al., 2014; Miralles et al., 2014a; Zhang et al., 2016). This study makes use of the latest version available, v4.0 It is a 37-year dataset spanning from 1980 to 2016 and is derived from a variety of sources, such as vegetation optical depth (VOD) and snow water

equivalents (SWEs), satellite-derived SM estimates, reanalysis air temperature and radiation, as well as a multi source precipitation product (Martens et al., 2017). It is available at a spatial resolution of 0.25°. A full description of the dataset, including an extensive validation using measurements from 64 eddy-covariance towers worldwide is provided by Martens et al. (2017). As for LAI, correlation and root mean squared differences are the two performance metrics used to

evaluate the representation of evapotranspiration from the three datasets.

The final product used in this study is a daily Gross Primary Production (GPP) estimate from, the FLUXCOM project (Jung et al., 2017). It is an upscaled product derived from the FLUXNET network. In FLUXCOM selected machine learning-based regression tools that span the full range of commonly applied algorithms (from model tree ensembles, multiple adaptive regression splines,

artificial neural networks, to kernel methods), with several representatives of each family are used to provide a spatial upscaling of GPP at regional to global scales. It is limited to a 0.5° spatial resolution and a daily temporal resolution over 1982–2013 (Tramontana et al., 2016). FLUXCOM fluxes can be used as a way of benchmarking LSMs on large scales (Jung et al., 2009, 2010; Beer et al., 2010; Bonan et al., 2011; Jung et al., 2011; Slevin et al., 2017). Product can be found in the Max

Planck    Institute    for    Biogeochemistry    Data    Portal    (https://www.bgc-jena.mpg.de/geodb/projects/Home.php). Correlation and root mean squared differences are the two performance metrics used to evaluate the representation of carbon uptake from the three datasets.

   3. **Results**





Averaged time-series of the six main land surface variables evaluated in this study over the whole
domain for 2010-2016 are illustrated on figure 1, they are (fig1.a) river discharge (although
averaging this variable over the whole domain has no real meaning, it is certainly useful to
appreciate the differences between the three data set), (fig1.b) snow depth, (fig1.c) leaf area index,
(fig1.d) liquid soil moisture in the second layer of soil (1-4 cm depth), (fig1.e) evapotranspiration
and (fig1.f) gross primary production. Land surface variables simulated with the ISBA LSM forced
by ERA-Interim (ei_S) are in blue, by ERA-5 (e5_S) with precipitation from ERA-Interim (e5ei_S)
in green and by ERA-5 in red. From figure 1 one can see that river discharge, snow depth and
surface soil moisture are the most impacted by the use of ERA-5, suggesting that precipitation is the
main driver of the differences.

### 3.1.    *Evaluations using in situ measurements*

This section presents the results of the comparison versus in situ observations of land surface
variables from model simulations using either ei_S, e5ei_S or e5_S starting with soil moisture. The
statistical scores for 2010–16 surface soil moisture from ei_S, e5ei_S and e5_S are presented in
Table II. Median R values on volumetric time-series (anomaly time series) are 0.66 (0.53), 0.69
(0.54) and 0.71 (0.58) while median ubRMSD are 0.052, 0.052 and 0.050 for ei_S, e5ei_S and and
e5_S, respectively. These results underline the better capability of the ISBA LSM to represent
surface soil moisture variability when forced by ERA-5 reanalysis. Also the latest configuration
(e5_S) presents more stations with better R values on volumetric time-series (anomaly time series)
than both ei_S and e5ei; respectively 60% and 75% (out of 110 and 107 stations, respectively). This
is also reflected on figure 2 illustrating correlations values on volumetric time-series (fig.2a) and
anomaly time-series (fig.2b) on maps. Stars symbols represent stations for which ISBA LSM
performs best when forced ERA-Interim, circles when it is forced by ERA-5 with ERA-Interim
precipitations and downward pointing triangles when it is forced by all ERA-5 atmospheric
variables. Both maps on figure 2 are dominated by downward pointing triangles.

172 out of 344 gauging stations retained for the evaluation according to the criteria described in the
methodology section presents efficiency scores in the [-1, 1] interval. Figure 3 represents
performance of each dataset for this pool of stations. Fig3.a is a scatterplot of efficiency scores
between in situ and simulated river discharges $Q$; efficiency scores for $Q$ simulated with either
ERA-5 but ERA-Interim precipitations (e5ei_S, green crosses) or ERA-5 (e5_S, red dots) function
of efficiency scores for $Q$ simulated using ERA-Interim (ei_S). When considering e5_S, almost all
the red dots are above the 1:1 diagonal suggesting a general improvement from the use of e5_S. For
a large part, e5ei_S green crosses are above this diagonal, suggesting that the improvement in e5_S
does not only comes from precipitation but from other variables, also. Median efficiency values are





0.06, 0.12 and 0.24 for ei_S, e5ei_S and e5_S, respectively. Fig.3b shows an histogram of river discharges ratio for ei_S (Qr_ei in blues), e5ei_S (Qr_e5ei in green) and e5_S (Qr_e5 in red),

median values are 0.67, 075 and 0.77, respectively. While all three experiments underestimate $Q$ (a value of 1 being a perfect match), the use of e5ei_S and e5_S leads to better results. Finally, figure 3c illustrates hydrographs for a river station in Lousiana (33.08°N, 1.52°W) representing scaled $Q$ (using either observed or simulated drainage areas), in situ data (black crosses), simulated river discharges from ei_S (blue solid line), e5ei_S (green solid line) and e5_S (red solid line). From this

hydrograph, the added value of e5_S is clear, particularly for the 2011 and 2015 main events. Efficiency scores are 0.47, 0.61 and 0.76 for ei_S, e5ei_S and e5_S, respectively. Figure 4 illustrates the added value of using e5_S (a) or e5ei_S (b) with respect to ei_S. For 156 out of the pool of 172 stations $NIC_{NSE}$ values computed using e5_S with respect to ei_S are positive (large blue circles) showing an general improvement from the use of e5_S (representing 91% of the

stations) with a median $NIC_{NSE}$ value of 14%. When considering e5ei_S versus ei_S, they are still 118 (69%) with a median $NIC_{NSE}$ value of 4% suggesting that the improvement in e5_S does not only comes from precipitation but from other variables, also. It is also worth-noticing that stations where a score degradation is observed (large red circles) are located in areas known for irrigation which is not represented in ISBA. All scores computed for seasons (December-January-Februray,

March-April-May, Jun-July-August, September-October-November) suggest the same ranking (not shown).

The mean snow depth bias of ei_S (see Figure 5) highlights a clear underestimation of winter snow depth accumulation mainly over the Rocky Mountains. This is likely a result of the underestimation of snowfall by ei_S associated with an overestimation of snow melt due to the coarse resolution of

the ei_S reflected in a smooth topography. The replacement of all forcing variables by e5_S but keeping ei_S precipitation (e5ei_S, Fig.5b) shows a slight increase in snow depth. This result justifies the above hypothesis that part of the snow underestimation is also due to temperature issues linked with a coarse model orography. Moving to the full e5_S forcing there is a clear increase of snow depth, when compared with both ei_S and e5ei_S forced simulations resulting from an

increase in snowfall in e5_S. In addition to the added values of e5_S in terms of the mean snow depth, the temporal variability and random errors are also improved (see Figure 6). Comparably with what was discussed for the mean bias, e5ei_S shows some benefits when compared with ei_S in terms of mae ($N_{MAE}$~5%), ubRMSD ($N_{ubRMSD}$~4%) and correlation ($NIC_R$ of 0.1) while e5_S has a clear improvement in mae ($N_{MAE}$~16%), ubRMSD ($N_{ubRMSD}$~14%) and correlation ($NIC_R$ of 0.25).

The improvements on the snow depth simulations are consistent throughout the entire snow covered



season (see Fig.6d) with a maximum improvement from January to March. These results highlight the cumulative effect of the forcing quality on the snow depth simulation.

Results from the comparisons between ei_S, e5ei_S, e5_S and in situ sensible and latent flux measurements are presented in table III and illustrated by figure 7 and 8. 37 stations present

significant correlation values (at p-value < 0.05). For sensible heat flux, median correlation and ubRMSD values are 0.62, 0.62 and 0.65, 34.85 $W.m^{-2}$, 30.66 $W.m^{-2}$ and 30.38 $W.m^{-2}$ for ei_S, e5ei_S and e5_S, respectively. For latent heat flux, they are 0.63, 0.62 and 0.70, 33.93 $W.m^{-2}$, 31.66 $W.m^{-2}$ and 30.98$W.m^{-2}$. As for surface soil moisture, river discharge and snow depth, e5_S presents better results than e5ei_S and ei_S. At the station level, figure 7 illustrates scatter plots of correlations and

ubRMSD for sensible and latent heat flux from ei_S, e5ei_S, e5_S against in situ measurements of sensible (fig.7a for correlation, fig.7c for ubRMSD) and latent (fig.7b for correlation, fig.7d for ubRMSD) heat flux. Scores for either e5ei_S (green dots) or e5_S (in red) are presented function of those for ei_S. When looking at the correlations, almost all of e5_S and e5ei_S symbols (in red and green, respectively on fig.7a, fig.7c) are above the 1:1 diagonal indicating that e5_S and e5ei_S

better represent sensible and latent heat flux than ei_S. Same tendency is observed for ubRMSD with most of the symbol below the 1:1 diagonal. If ubRMSD values are comparable for e5_S and e5ei_S, R values are clearly higher for e5_S. Finally figure 8 presents NIC scores based on correlations values between in situ measurements from the fluxnet sites data and (fig8.a) e5_S with respect to ei_S for sensible heat flux and (fig8.b) e5_S with respect to ei_S for latent heat flux.

Normalised ubRMSD values between in situ measurements from the fluxnet sites data and (fig8.c) e5_S with respect to ei_S for sensible heat flux and (fig8.d) e5_S with respect to ei_S for latent heat flux. Blue circles indicate improvement compared to ei_S (positive values of $NIC_R$ in fig.8a,b, and negative values of $N_{ubRMSD}$ in fig.8c,d) while red circles correspond to a degradation (negative values of $NIC_R$ in fig.8a,b, and positive values of $N_{ubRMSD}$ in fig.8c,d). The four maps of figure 8 are largely

dominated by large blue circles, therefore with dominant improvements.

### 3.2. *Evaluations using satellite derived estimates*

Figure 9 illustrates the comparison between ESA CCI SSM_v4 and soil moisture from ISBA second layer of soil over 2010-2016. Fig.9a shows seasonal correlations on volumetric time-series and fig9.b on anomaly time-series. Scores for ISBA LSM forced by ERA-Interim (ei_S) are in blue,

ERA-5 but with precipitation from ERA-Interim (e5ei_S) green and ERA-5 (e5_S) in red. From fig9.a one can appreciate the added value of using ERA-5 atmospheric forcing particularly from Aprils to September. It is also interesting to notice that when using all ERA-5 atmospheric fields except for the precipitations, a similar added value is noticeable suggesting that all improvements from ERA-5 do not only come from precipitation. However when evaluating the short-term



variability of soil moisture (i.e. removing the seasonal effect), it is really ERA-5 that provides the best results. Correlation on volumetric (anomaly) time-series for all grid points put together over 2010-2016 are 0.668 (0.464), 0.682 (0.468) and 0.689 (0.490) for ei_S, e5ei_S and e5_S, respectively. Additionally to the global seasonal scores, fig.9c and fig.9d present maps of correlations differences between soil moisture from e5_S and ei_S on volumetric time-series and

anomaly time-series, respectively. Grey areas represent areas that were flagged out for elevation greater than 1500 m above sea level. As visible on fig.9c and fig.9d the use of ERA-5 ,mainly leads to improvements all over the considered domain. When applying Eq.3 (NIC) to $R_{e5}$ with respect to $R_{ei}$ on volumetric (anomaly) time-series, 68% (77%) of the values are positives (indicating an improvement from e5) with median values of 4.5% (4.11%) and include values up to 40% (45%). It

shows the added value of using ERA-5 to force ISBA LSM compared to ERA-Interim.

Figure 10 illustrates seasonal scores between ISBA LSM forced by either ERA-Interim (ei_S in blue) ERA-5 but ERA-Interim precipitation (e5ei in green) or ERA-5 (e5_S in red) and the three lasting dataset; (fig10.a, fig10.b) evapotranspiration estimates from the GLEAM project over 2010-2016, (fig10.c, fig10.d) upscaled GPP from the FLUXCOM project over 2010-2013 and (fig10.e,

fig10.f) LAI estimates from the Copernicus GLS project over 2010-2016. Left column (fig10.a, c and e) are for RMSD and right column (fig8.b, d, e) for correlations. For evapotranspiration and to a lesser extend GPP, one can notice a decrease in RMSD when using ERA-5 atmospheric reanalysis compared to ERA-Interim atmospheric reanalysis. However it fails at improving LAI. Considering evapotranspiration, correlation (RMSD) values for all grid points put together over 2010-2016 are

0.786 (0.927 kg.m$^{-2}$.d$^{-1}$), 0.778 (0.917 kg.m$^{-2}$.d$^{-1}$) and 0.795 (0.889 kg.m$^{-2}$.d$^{-1}$) for ei_S, e5ei_S and e5_S, respectively. They are 0.726 ( 2.429 kg.m$^{-2}$.d$^{-1}$), 0.733 (2.167 kg.m$^{-2}$.d$^{-1}$) and 0.734 (2.227 kg.m$^{-2}$.d$^{-1}$) for GPP and 0.715 (1.050 m$^{2}$.m$^{-2}$), 0.710 (1.026 m$^{2}$.m$^{-2}$) and 0.697 (1.079 m$^{2}$.m$^{-2}$) for LAI. Improvements (in red) and degradations (in blue) from the use of ERA-5 in the ISBA LSM with respect to ERA-Interim for evapotranspiration, Gross Primary Production and Leaf Area Index

are illustrated by figure 11 (respectively from top to bottom). Fig.11a, c and e follow eq.(7) ($N_{RMSD}$) while Fig.11b, d and f follow Eq.(3) ($NIC_R$). Both $N_{RMSD}$ and $NIC_R$ suggest an improvement from the use of ERA-5 as the two figures are mainly dominated by red colors, they represent 56% and 53% of the domain, respectively for evapotranspiration (fig.11a, b), 60% and 69% for GPP (fig.11c, d) but only 47% and 44% for LAI (fig.11e, f). On figure 11, values between -12.5 and 12.5 for

$N_{RMSD}$, -25 and 25 $NIC_R$ are not represented for sack of clarity.

   4.  **Discussion and conclusions**

This study assesses the ability of ECMWF recently released ERA-5 atmospheric reanalysis to force the ISBA LSM with respect to ERA-Interim reanalysis over North America for 2010-2016. The



results presented above using the three atmospheric reanalysis data set (ERA-Interim -ei_S-, ERA-5
but with precipitation from ERA-Interim -e5ei_S- and ERA-5 -e5_S- all meteorological variables)
to force the ISBA LSM provide two important insights ; (i) firstly the use of ERA-5 leads to
significant improvements in the representation of the LSVs linked to the terrestrial water cycle
assessed in this study (surface soil moisture, river discharges, snow depth and turbulent fluxes) but
failed impacting LSVs linked to the vegetation cycle (carbon uptake and LAI). (ii) Secondly if most
of the improvements seems to come from a better representation of the precipitation in ERA-5, the
e5ei_S experiment also present improvements with respect to the ei_S experiment suggests that the
other meteorological forcing from ERA-5 are better represented too. It is however acknowledge that
the use of 3-hourly ERA-Interim liquid and solid precipitations re-scaled at an hourly time step in
ERA-5 might have sometimes led to inconsistent configurations (e.g., precipitations while having a
very strong net radiation). Albergel et al., 2017, 2018 (in prep.) recently presented a Land Data
Assimilation System (LDAS-Monde) able to sequentially assimilate satellite derived estimates of
surface soil moisture and LAI. They found that the main improvements of their analysis (i.e. with
assimilation) when compared to an open-loop experiment (simple model run) were linked to
vegetation variables and the assimilation of vegetation estimates. They have also proposed further
advances on a better use of satellite-based microwave data in the assimilation system. Having
LDAS-Monde analysis forced by ERA-5 atmospheric forcing should both combined the strengths
of an improved atmospheric reanalysis on the terrestrial water cycle and of the assimilation of
satellite derived products on the vegetation cycle. Effort will now be concentrate on the use of
ERA-5 and strengthening LDAS-Monde through the direct assimilation of satellite-based soil
moisture and vegetation properties from microwave remote sensing. It will enable fostering links
with potential applications like climate reanalysis of the LSVs as well as going from a monitoring
system of the LSVs and extreme events (like agricultural drought) to a forecasting system.
Preliminary results suggest that a LSVs forecast initialized by an analysis is more robust than one
initialized by a simple model run (Albergel et al., 2018, in prep). Preliminary tests over Europe also
indicate similar benefits from the use of ERA-5 (not shown). When the whole ERA-5 period will be
available (1979-present), in addition with the availability of the ERA-5 10-member Ensemble of
Data Assimilation (at lower spatial and temporal resolution though), it will be possible to develop a
global long term ensemble of land surface variables reanalysis forced by high quality atmospheric
data. It will make it possible providing uncertainties in the representation of the atmospheric
forcing, while land surface variables may require special considerations and perturbation methods.
Capturing those uncertainties coming from the simplifications and assumptions in the LSM is of
paramount interest for many applications from monitoring to forecasting.



**Acknowledgments-***Results where Generated using Copernicus Climate Change Service Information 2017. E. Dutra work was supported by the Portuguese Science Foundation (FCT)*
*under project IF/00817/2015.*





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




## Tables and Illustrations

Table I : Evaluation datasets and associated metrics used in this study.

| Datasets used for the evaluation | Source | Metrics associated |
|---|---|---|
| In situ measurements of soil moisture (USCRN, Bell et al., 2013) | https://www.ncdc.noaa.gov/crn | R (on both volumetric and anomaly time-series) ubRMSD |
| In situ measurements of streamflow (USGS) | https://nwis.waterdata.usgs.gov/nwis | Nash Efficiency (NSE), Normalized Information Contribution (NIC) based on NSE, Ratio of simulated and observed streamflow (Q) |
| In situ measurements of snow depth (GHCN, Menne et al., 2012a, b) | https://www.ncdc.noaa.gov/climate-monitoring/ | R, stde and MAE, NIC based on R, Normalized stde and MAE |
| In situ measurements of sensible and latent heat fluxes (FLUXNET-2015) | http://fluxnet.fluxdata.org/data/fluxnet2015-dataset/ | R, ubRMSD, NIC based on R, Normalized ubRMSD |
| Satellite derived surface soil moisture (ESA CCI SSM v4, Dorigo et a., 2015, 2017) | http://www.esa-soilmoisture-cci.org | R (on both volumetric and anomaly time-series) |
| Satellite derived Leaf Area Index (GEOV1, Baret et al., 2013) | http://land.copernicus.eu/global/ | R and RMSD, NIC based on R, Normalized RMSD |
| Satellite-driven model estimates of land evapotranspiration (GLEAM, Martns et al., 2017) | http://www.gleam.eu | R and RMSD, NIC based on R, Normalized RMSD |
| Upscaled estimates of Gross Primary Production (GPP, Jung et al., 2017) | https://www.bgc-jenna.mpg.de/geodb/projects/Home.php | R and RMSD, NIC based on R, Normalized RMSD |

**Table II: Comparison of surface soil moisture with in situ observations for ei_S, e5ei_S and e5_S over 2010-2016 (April to September months are considered). Median correlations R (on volumetric and anomaly time series) and ubRMSD are given for the USCRN. Scores are given for significant correlations with p-values <0.05.**

| | Median R* on volumetric time series ( % of stations for which this configuration is the best) | Median R** on anomalies time series ( % of stations for which this configuration is the best) | Median ubRMSD* ($m^3 m^{-3}$) ( % of stations for which this configuration is the best) |
|---|---|---|---|
| ei_S | 0.66 (20 %) | 0.53 (15 %) | 0.052 (19 %) |
| e5ei_S | 0.69 (20 %) | 0.54 (10 %) | 0.052 (24 %) |
| e5_S | 0.71 (60 %) | 0.58 (75 %) | 0.050 (57 %) |

**\* only for stations presenting significant R values on volumetric time series (p-value<0.05): 110 stations**
**\*\* only for stations presenting significant R values on anomaly time series (p-value<0.05): 107 stations**





**Table III: Comparison of sensible (H) and latent (LE) heat flux with in situ observations for ei_S, e5ei_S and e5_S. Median correlations (R) and median ubRMSD are given for the fluxnet stations. Scores are given for significant correlations with p-values <0.05.**

|  | H Median R* ( % of stations for which this configuration is the best) | H Median ubRMSD* W.m⁻² ( % of stations for which this configuration is the best) | LE Median R* ( % of stations for which this configuration is the best) | LE Median ubRMSD* W.m⁻² ( % of stations for which this configuration is the best) |
|---|---|---|---|---|
| ei_S | 0.62 (8 %) | 34.85 (0 %) | 0.63 (8 %) | 33.93 (14 %) |
| e5ei_S | 0.62(27 %) | 30.66 (22 %) | 0.62 (11 %) | 31.66 (24 %) |
| e5_S | 0.65 (65 %) | 30.38 (78 %) | 0.70 (81 %) | 30.98 (62 %) |

**\* only for stations presenting significant R values (p-value<0.05): 37 stations**





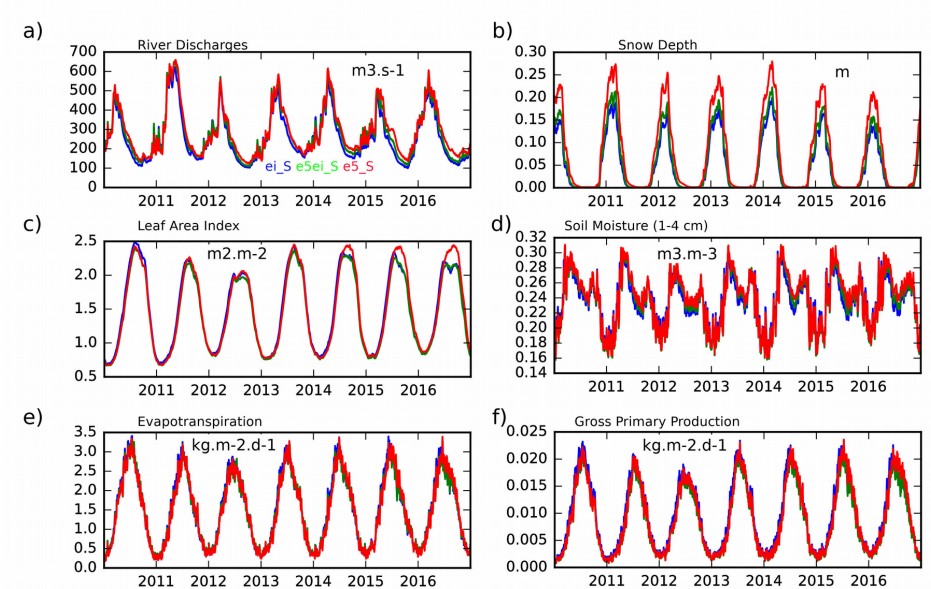

**Figure 1 : Averaged time-series of the 6 main land surface variables evaluated in this study over the whole domain for 2010-2016. (a) river discharge, (b) snow depth, (c) leaf area index , (d) liquid soil moisture in the second layer of soil (1-4 cm depth), (e) evapotranspiration and (f) gross primary production. Land surface variables simulated with SURFEX forced by ERA-Interim (ei_S) are in blue, by ERA-5 (e5_S) with precipitation from ERA-Interim (e5ei_S) in blue and by ERA-5 in red.**

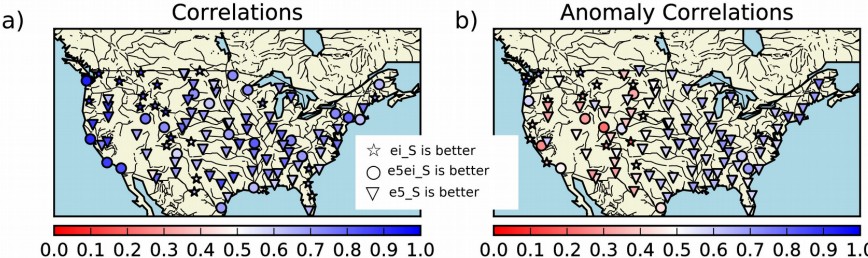

**Figure 2 : Maps of correlation (R) on volumetric time-series (a) and anomaly time-series (b) between in situ measurements at 5 cm depth from the USCRN network and the ISBA Land Surface Model within the SURFEX modeling platform forced by either ERA-Interim (ei_S), ERA-5 with ERA-Interim precipitations (e5ei_S) and ERA-5 (e5_S). For each stations presenting significant R (p-values < 0.05) simulation that presents the better R values is represented. Stars symbols are when ei_S, presents the best value, circles when it e5ei_S and downward pointing triangles when it is e5_S.**



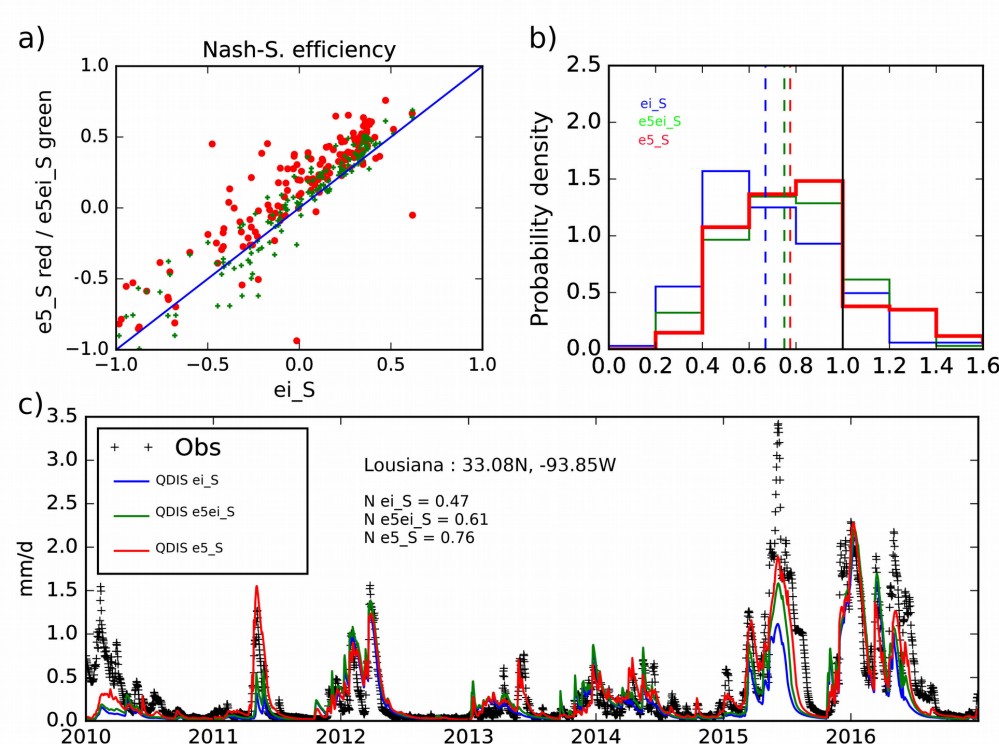

**Figure 3: (a) Scatterplots of efficiency scores between in situ and simulated river discharges *Q*; efficiency scores for *Q* simulated with SURFEX forced either by ERA-5 but ERA-Interim precipitations (e5ei_S, green crosses) or ERA-5 (e5_S, red dots) function of efficiency scores for *Q* simulated using ERA-Interim (ei_S). (b) Histograms of river discharges ratio for ei_S (Qr_ei in blues),**
**e5ei_S (Qr_e5ei in green) and e5_S (Qr_e5 in red). (c) Hydrograph for a river station in Lousiana (33.08°N, 1.52°W) representing scaled *Q* (using either observed or simulated drainage areas), in situ data (black crosses), simulated river discharges from ei_S (blue solid line), e5ei_S (green solid line) and e5_S (red solid line).**





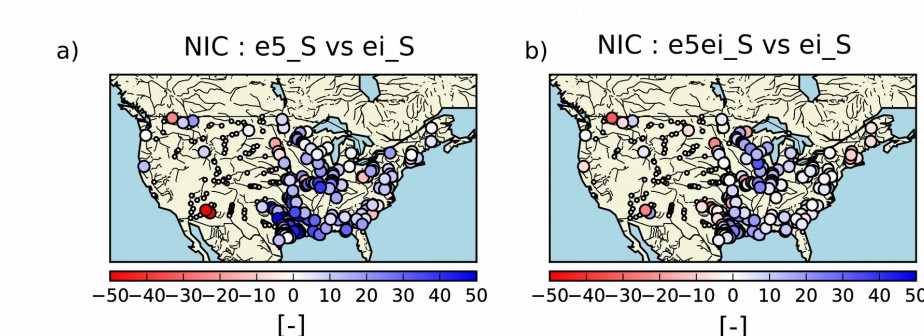

**Figure 4: Normalized Information Contribution scores based on efficiency scores (NIC$_{NSE}$) (a) e5_S with respect to ei_S and (b) e5ei_S with respect to ei_S. Small dots represent station for which the benchmark experiment (ei_S) present efficiency scores smaller than -1, large circles when it presents values higher than -1. Positive values (blue large circles) suggest an improvement over ei_S, negative values (red large circles) a degradation. For sack of clarity, a factor 100 has been applied on NIC.**

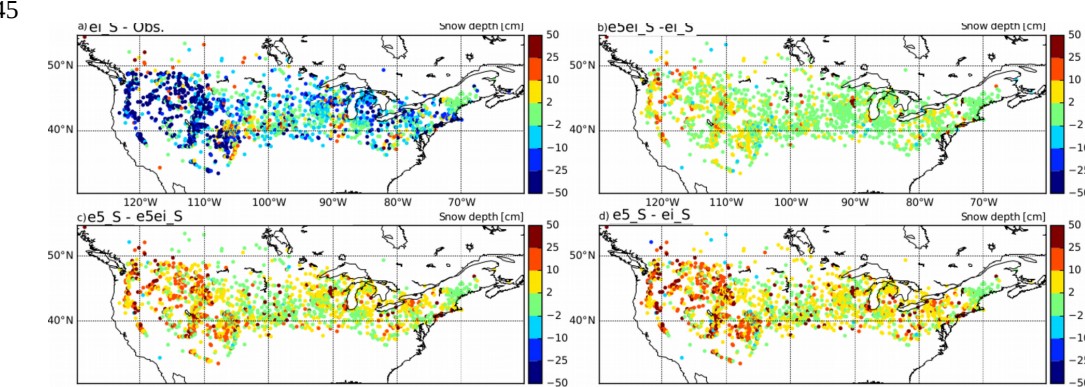

**Figure 5: Mean snow depth bias for December-January-February in ei_S (a) and differences between e5ei_S and ei_S (b), e5_S and e5ei_S (c), e5_S and ei_5 (d).**



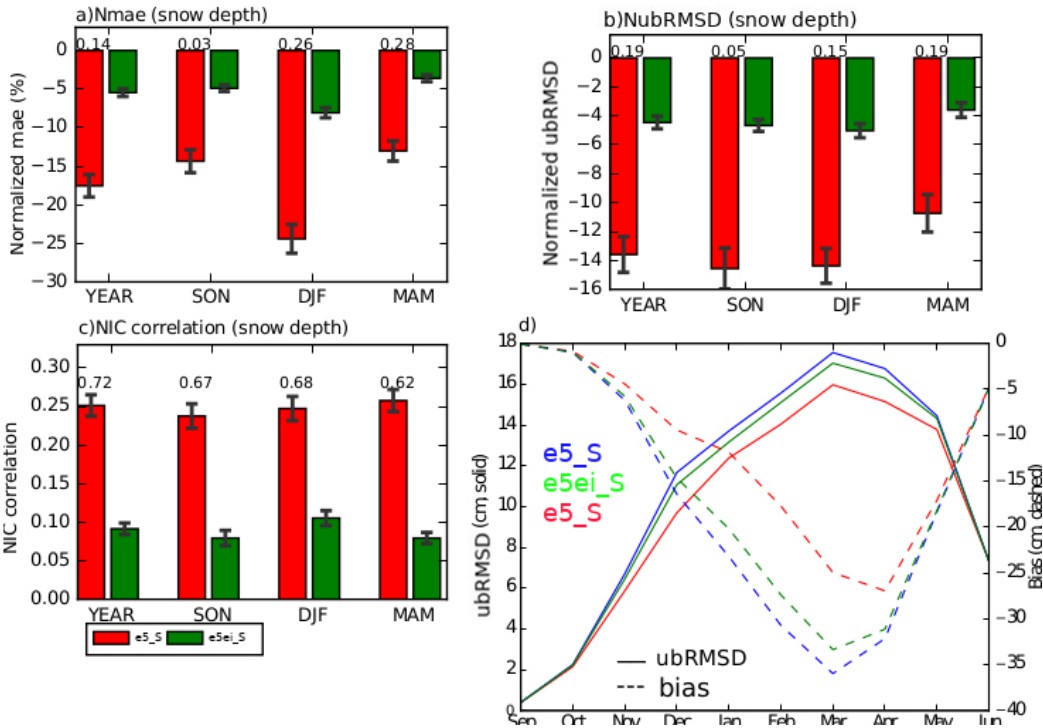

**Figure 6: Snow depth Normalized Information Contribution scores (eq. 3) in respect to ei_S for (a) the mean absolute error (MAE), (b) unbiased Root Mean Square Differences (ubRMSD) and (c) correlation. The error bars denote a 95% confidence interval of the mean derived from a 10000 samples bootstrapping. (b) shows the mean annual cycle of the bias (solid lines) and ubRMSD (dashed lines) averaged over all stations.**





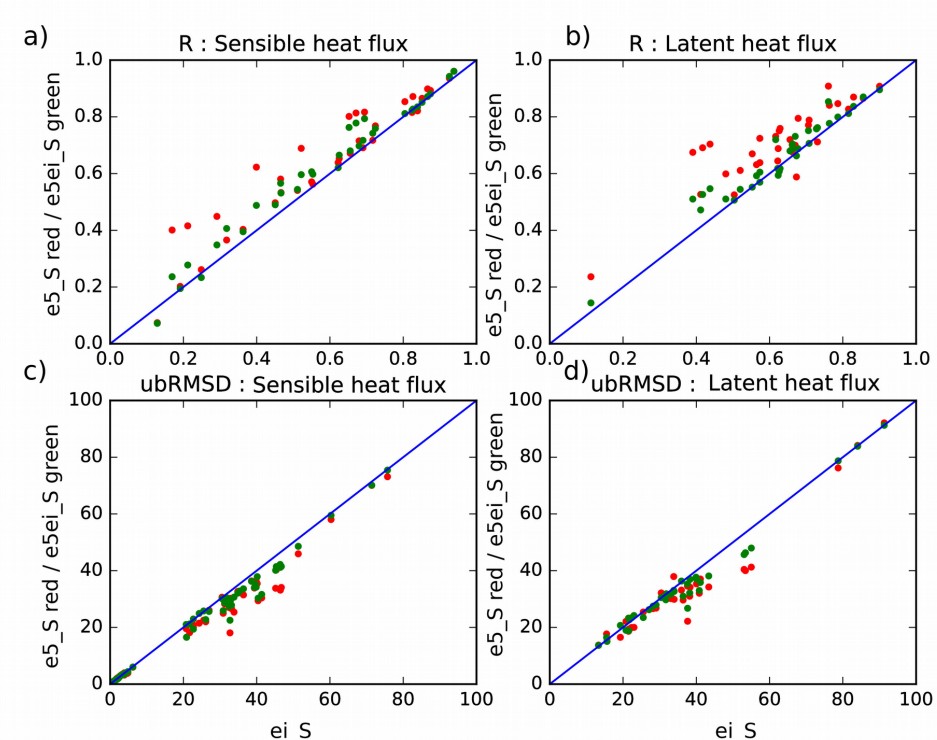

**Figure 7: scatterplots illustrating evaluation of ei_S, e5ei_S, e5_S against in situ measurements sensible (a for correlation, c for ubRMSD) and latent (b for correlation, d for ubRMSD) heat flux. Scores for either e5ei_S (green dots) or e5_S (in red) are presented function of those for ei_S.**



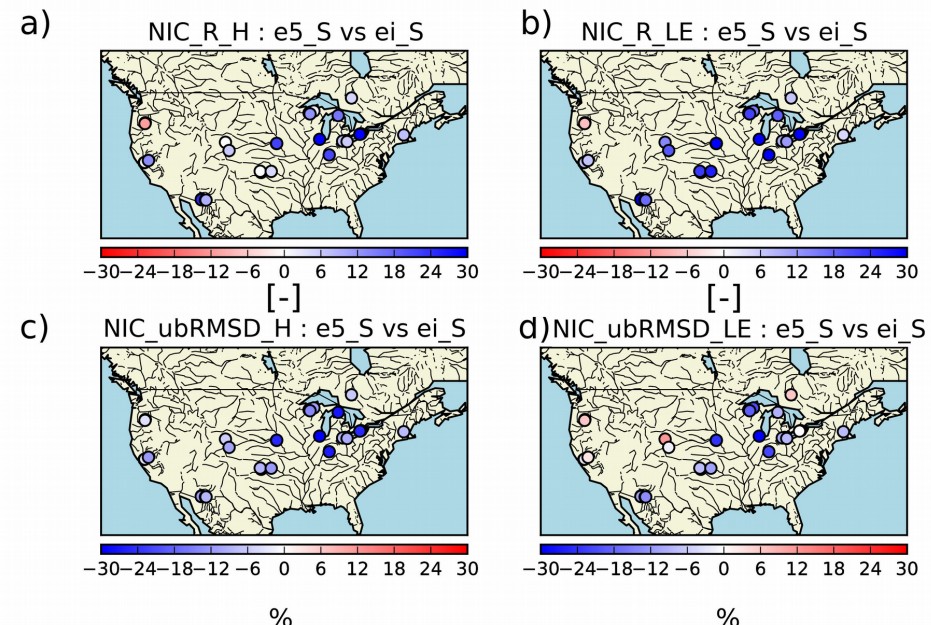

**Figure 8: Normalized Information Contribution scores based on correlations values between in situ measurements from the fluxnet sites data and (a) e5_S with respect to ei_S for sensible heat flux and (b) e5_S with respect to ei_S for latent heat flux. Normalised ubRMSD values between in situ measurements from the fluxnet sites data and (c) e5_S with respect to ei_S for sensible heat flux and (d) e5_S with respect to ei_S for latent heat flux. Blue circles indicate improvement compared to ei_S (positive values of $NIC_R$ in a,b, and negative values of $N_{ubRMSD}$ in c,d) whereas red circles correspond to a degradation (negative values of $NIC_R$ in a,b, and positive values of $N_{ubRMSD}$ in c,d). For sack of clarity a factor 100 has been applied on $NIC_R$.**





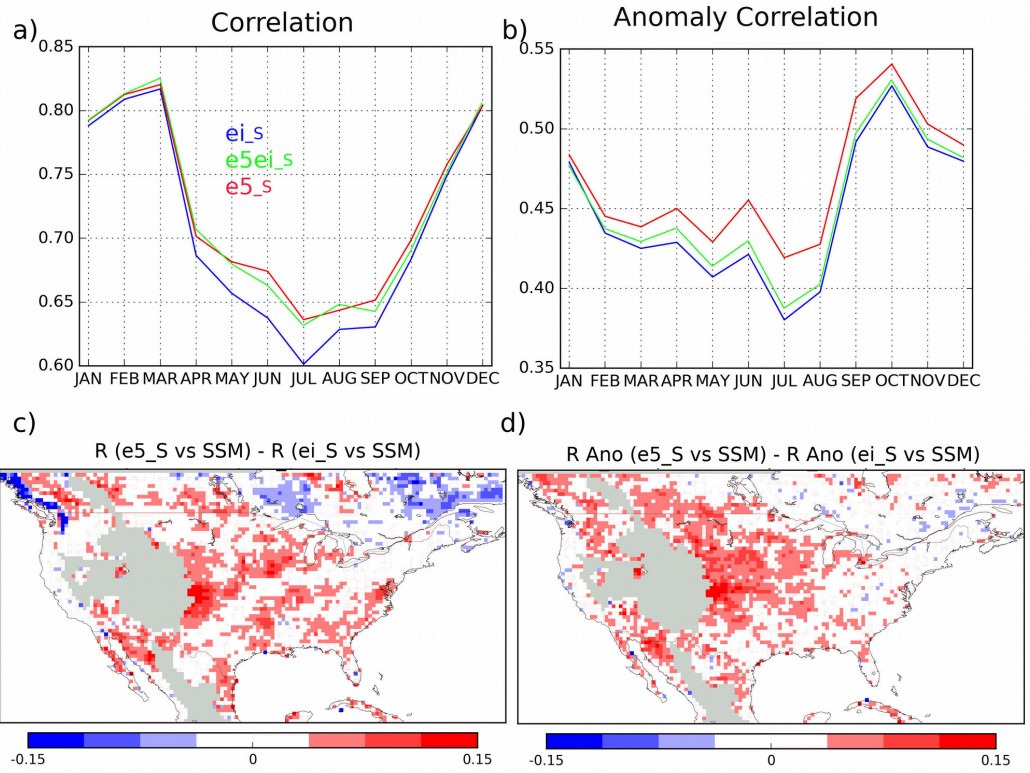

**Figure 9 : Seasonal correlations on (a) volumetric time-series and (b) anomaly time-series between surface soil moisture estimates from the ESA CCI project (ESA-CCI SSM v4) and soil moisture from the second layer of soil of the ISBA LSM forced by ERA-Interim (ei_S in blue), ERA-5 but with precipitation from ERA-Interim (e5ei_S in green) and ERA-5 (e5_S in red) over 2010-2016. Maps of correlations differences between soil moisture from e5_S and ei_S on volumetric time-series (c) and anomaly time-series (d), areas in red represent an improvement from the use of ERA-5. Grey areas represents areas that where flagged out for elevation greater than 1500 m above sea level.**





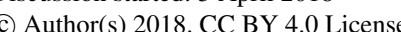

Figure 10: Seasonal scores between ISBA LSM within SURFEX forced by either ERA-Interim (ei_S in blue) ERA-5 but ERA-Interim precipitation (e5ei_S in green) or ERA-5 (e5_S in red) and (a, b) evapotranspiration estimates from the GLEAM project over 2010-2016, (c, d) upscaled GPP from the FLUXCOM project over 2010-2013 and (e, f) LAI estimates from the Copernicus GLS project over 2010-2016. Left column (a, c and e) are for RMSD and right column (b, d, e) for correlations.



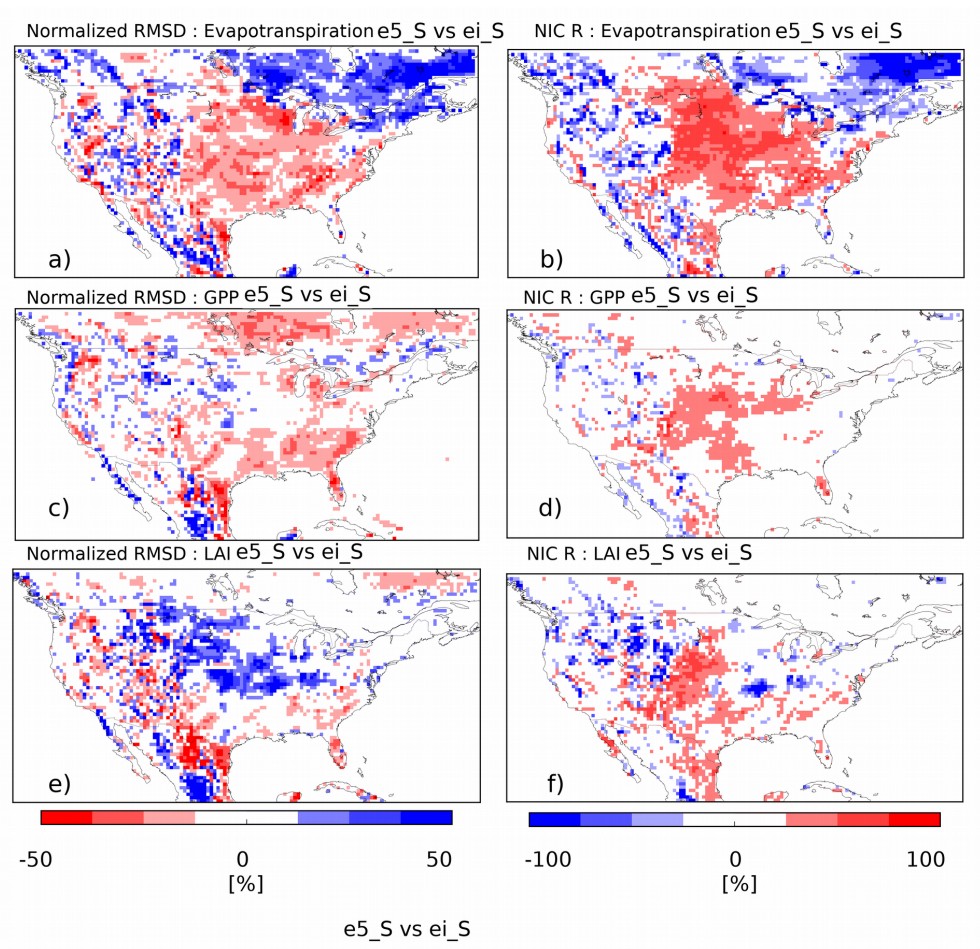

**Figure 11: Normalised RMSD (a, c, e) and Normalised Information Contribution (NIC, b, d, f) based on correlation values for e5_S simulations with respect to ei_S simulations for three land surface variables; evapotranspiration, Gross Primary Production and Leaf Area Index from top to bottom. Areas in red represent an improvement from the use of ERA-5. For sack of clarity a factor 100 has been applied on NIC$_R$.**
