# Peer review of "ERA-5 and ERA-Interim driven ISBA land surface model simulations : Which one performs better?"

_Hydrology and Earth System Sciences, 2018_

## Referee Comment (RC1) · W. Wagner (Referee) · 24 Apr 2018

GENERAL COMMENTS

This is an excellent paper, so I keep my review short.

This study is one of the first evaluations of ECMWF's latest reanalysis ERA-5. ERA-5 brings a number of important improvements over the widely used ERA-Interim reanalysis, including more detailed process descriptions and finer spatial- and temporal resolution. The authors evaluate ERA-5 by forcing a land surface model with ERA-Interim and ERA-5 data. To single out the effects of the enhanced precipitation estimates in

ERA-5, the authors add a third forcing data set composed of ERA-5 data with only precipitation coming from ERA-Interim. The authors comprehensively evaluate the model simulations over North America with an impressive number of reference data, from in situ networks (runoff, soil moisture, evaporation, snow depth..) and remote sensing (soil moisture, LAI, ...). The results are realistic, pointing to consistent improvements of ERA-5 over ERA-Intermin in particular for the hydrologic components.

SPECIFIC COMMENTS

Lines 254: Please explain, why discarding stations with drainage areas differing by more than 20 % from the simulated one makes sense.

Section 4: This section is a bit weak. Consider e.g. to summarize key findings with respect to each of the analysed processes and data sets. I also think that too much of the present discussion deals with other on-going work of the authors. Please try to discuss relevant links with similar undertakings currently on-going in the US, Japan, China, etc..

Albergel et al. (2018) not in the references

Figure 4: "For sake of clarity . . ."

---

## Referee Comment (RC2) · Anonymous Referee #2 · 3 May 2018

This paper documents the improvements in land surface simulations driven by ERA-5 relative to ERA-Interim. The work is relevant from the standpoint of documenting the improvements in ERA-5 as it relates to hydrology. The manuscript is somewhat sloppily prepared (a figure was missing!). My comments and suggestions are below.

Major comments: 1. The text has numerous language issues and grammar mistakes, some of which are listed below. I ran out of steam documenting all of them. I assume that the author would take a fresh and careful look at the manuscript to correct them all (including the ones that are not listed). Note that Figure 5 was missing in the manuscript version that I reviewed. 2. The paper is really an offline LDAS simulation,

[Figure]

but makes no mention of other LDAS work. The literature review should encompass the recent work in this regard that have considered the assimilation of land measurements (NCA-LDAS, for.e.g). I think the paper would be more powerful and of broader interest if the authors can document how the ERA-5 forced system compares with the such LDAS efforts. For example, does ERA-5 have comparable skills to NLDAS2, the defacto standard land product over CONUS? How does it compare to MERRA2 and ERA-Interim-Land? Without such comparisons, the paper sounds more as a technical report the impact of ERA related changes. 3. I found the use of metrics to be a bit convoluted and inconsistent. For example, why use NIC for metrics such as R, instead of simply taking a difference? NIC is more useful when the dynamic range of the metrics are really large (NSE ranges from -infinity to 1, so a large negative value would blow up a domain average). I suggest sticking with simple differences so that the impact on the model runs are more intuitive. Why use ubRMSED to look at impacts on fluxes? (It is used for soil moisture because of the large climatological differences)Also, I would change the sign of N_MAE and N_ubRMSD to be the same as that of NIC values (Positive value indicating improvements). 4. Section 3.1: I think the descriptions need to tone down the language on how much improvements are actually gained. From table 1, it looks like the improvements are quite small though they are systematic with the new version. I think it is important to quantify the magnitude of improvements (showing their spatial distribution through, for e.g., histograms).

Minor comments: 1. Fix the sentence starting with 'ERA-5 important changes ..' to something like ' ERA-5 has important changes relative to ERA-Interim former atmospheric reanalysis including . . .' 2. Change the sentence 'ERA-5 is forseen .. ' to something like 'As ERA-5 is expected to replace ERA-Interim reanalysis, this study assesses whether . . .' 3. Change the sentence 'ERA-5 impact on the ISBA ..' to 'ERA-5's impact on ISBA LSM relative to ERA-Interim is evaluated using remote sensing . . .' 4. Line 34 – Fix 'Interim ..' to 'Interim.' (only one period). 5. Line 36: change 'extend' to 'extent' 6. Line 46: Change 'essentials' to 'essential' 7. Line 52: Change 'progresses' to 'progress' 8. Line 55: Add a comma after 'decade'. 9. Lines 58-60: MERRA is

retired. More appropriate to refer to MERRA2 papers. Given that this paper focuses on land-only simulations, there should be a description of LDAS analysis forced by observed precipitation (and meteorology) such as NLDAS, GLDAS ,etc. 10. Lines 65-68: Similar to abstract, these sentences are awkwardly written. 11. Lines 96: Change to say 'Section 2 presents the details of two atmospheric ..' 12. Line 120: Change to say 'which allows it to use ..' 13. Line 132: Add a comma after 'study' 14. Section 2.3: I would say 'interpolated to' rather than 'interpolated at': What interpolation methods were used? 15. Line 217: Kumar et al. (2009) is not in the list of references. 16. Line 237: I would not say 'artificially increasing the perceived agreement' – Just that the skill values are higher because it includes the seasonal cycle. 17. Line 239: 'Monthly averaged are also computed' (?) 18. Line 240: Change 'week' to 'weeks'. 19. Lines 239-242: It sounds like this you are really computing the z-scores rather than anomalies, since you are scaling the differences with standard deviation 20. Line 245: What significance test is done to compute the p-values? This varies depending on the metric of interest. In particular, since several derived metrics (NICs) are used here, how did you compute the statistical significance? 21. Line 265: Change from 'an NSE' to 'a NSE' 22. Line 323: Change 'exercises' to 'studies' 23. Line 347: Change 'equivalents' to 'equivalent' 24. Figure 1: As the authors describe, this figure is not very useful. The lines are too close to each other in most part. It will be easier to see them if you plot the differences (relative to ei_S ; then you only have two lines). Another option is to show a seasonal cycle rather than the entire time series. 25. Lines 390-395: Say NSE rather than 'efficiency' 26. Lines 465: Change 'Aprils' to 'April' 27. Lines 479-480: What does 'lasting dataset' mean? 28. Lines 509-510: Fix – 'It is however acknowledge that ..'

---

## Author Response (AR1)

Response to Reviewer 1 are structured as follow: (1) 1.X: comments from Reviewer 1, (2) Response to 1.X: author's response and author's changes in manuscript when any. For sake of clarity, line and page from the first submission is used.

**Reviewer#1 : Wolfgang Wagner**

**General comments**
**This is an excellent paper, so I keep my review short.**
**This study is one of the first evaluations of ECMWF's latest reanalysis ERA-5. ERA-5 brings a number of important improvements over the widely used ERA-Interim reanalysis, including more detailed process descriptions and finer spatial- and temporal resolution. The authors evaluate ERA-5 by forcing a land surface model with ERA-Interim and ERA-5 data. To single out the effects of the enhanced precipitation estimates in ERA-5, the authors add a third forcing data set composed of ERA-5 data with only precipitation coming from ERA-Interim. The authors comprehensively evaluate the model simulations over North America with an impressive number of reference data, from in situ networks (runoff, soil moisture, evaporation, snow depth..) and remote sensing (soil moisture, LAI, ...). The results are realistic, pointing to consistent improvements of ERA-5 over ERA-Intermin in particular for the hydrologic components.**

Dear Wolfgang Wagner, many thanks for reviewing the manuscript and for highlighting its relevance and interest. Your comments and suggestion led to an improved version of the manuscript, in particular with respect to section 4.

**Specific comments**
**1.1 [Lines 254: Please explain, why discarding stations with drainage areas differing by more than 20 % from the simulated one makes sense.]**

Author's response to 1.1:
The following explanation has been added to the revised version of the manuscript:

P.8, L.254, "This criterion aims to ensure a meaningful comparison between observed and simulated values. It is necessary for coping with the significant distortions in the model representation of the river network that are caused by the coarse spatial resolution of the CTRIP global river network (0.5°x0.5°)."

**1.2 [Section 4: This section is a bit weak. Consider e.g. to summarize key findings with respect to each of the analysed processes and data sets. I also think that too much of the present discussion deals with other on-going work of the authors. Please try to discuss relevant links with similar undertakings currently on-going in the US, Japan,China, etc..]**

Author's response to 1.2
In agreement with comment 1.2, the revised version of the manuscript have the following modifications:

P.16, L.350-352: "Albergel et al., 2017, 2018 (in prep.) recently presented a Land Data Assimilation System (LDAS-Monde) able to sequentially assimilate satellite derived estimates of surface soil moisture and LAI."
**is now:**
"ERA-5 has a great potential to further improve the representation of land surface variables if used to force offline LDAS. In the past recent years, several LDAS have emerged at different spatial scales, (i) regional like the Coupled Land Vegetation LDAS (CLVLDAS, Sawada and Koike, 2014,

Sawada et al., 2015), the Famine Early Warning Systems Network (FEWSNET) LDAS (FLDAS, McNally et al., 2017), (ii) continental like the North American LDAS (NLDAS, Mitchell et al., 2004; Xia etal., 2012), the National Climate Assessment LDAS (NCA-LDAS Kumar et al., 2018) as well as at (iii) global scale like the Global Land Data assimilation (GLDAS, Rodell et al., 2004) and more recently LDAS-Monde (Albergel et al., 2017, 2018 in prep). LDAS-Monde is a global capacity system able to sequentially assimilate satellite derived estimates of surface soil moisture and LAI."

New references:

-Albergel, C., S. Munier, A. Bocher, C. Draper, D. J. Leroux, A. L. Barbu, J.-C. Calvet: LDAS-Monde global capacity integration of satellite derived observations applied over North America: assessment, limitations and perspectives. to be sumitted to Remote Sensing, Special Issue "Assimilation of Remote Sensing Data into Earth System Models", 2018

-Kumar, S.V., M. Jasinski, D. Mocko, M. Rodell, J. Borak, B. Li, H. Kato Beaudoing, and C.D. Peters-Lidard: NCA-LDAS land analysis: Development and performance of a multisensor, multivariate land data assimilation system for the National Climate Assessment. J. Hydrometeor., 0, https://doi.org/10.1175/JHM-D-17-0125.1

-McNally, A., Arsenault, K., Kumar, S., Shukla, S., Peterson, P., Wang, S., Funk, C., Peters-Lidard, C. D. and Verdin, J. P.: A land data assimilation system for sub-Saharan Africa food and water security applications. Scientific Data, 4, 170012, :10.1038/sdata.2017.12, 2017.

-Mitchell, K. E., et al. The multi-institution North American Land Data Assimilation System (NLDAS): Utilizing multiple GCIP products and partners in a continental distributed hydrological modeling system, J. Geophys. Res.,109, D07S90, 2004. doi:10.1029/2003JD003823

-Muñoz-Sabater, Joaquín, Emanuel Dutra, Gianpaolo Balsamo, Souhail Boussetta, Ervin Zsoter, Clement Albergel, Anna Agusti-Panareda: ERA5-Land: an improved version of the ERA5 reanalysis land component. Joint ISWG and LSA-SAF Workshop, 26-28 June 2018, Lisbon, Portugal.

-Rodell, M., P. R. Houser, U. Jambor, J. Gottschalck, K. Mitchell, C.-J. Meng, K. Arsenault, B. Cosgrove, J. Radakovich, M. Bosilovich, J. K. Entin, J. P. Walker, D. Lohmann, and D. Toll, The Global Land Data Assimilation System, Bull. Amer. Meteor. Soc., 85(3), 381–394, 2004.

-Sawada, Y., T. Koike, and J. P. Walker, A land data assimilation system for simultaneous simulation of soil moisture and vegetation dynamics, J. Geophys. Res.Atmos., 120, doi: 10.1002/2014JD022895, 2015.

-Sawada, Y., and T. Koike, Simultaneous estimation of both hydrological and ecological parameters in an ecohydrological model by assimilating microwave signal, J. Geophys. Res. Atmos., 119, doi:10.1002/2014JD021536, 2014.

Xia, Y., et al. 2012, Continental-scale water and energy flux analysis and validation for the North American Land Data Assimilation System project phase 2 (NLDAS-2): 1. Intercomparison and application of model products, J. Geophys. Res., 117, D03109, doi:*10.1029/2011JD016048*, 2012.

**1.3 [Albergel et al. (2018) not in the references]**

Author's response to 1.3: it is now corrected in the revised version of the manuscript.

**1.4 [Figure 4: "For sake of clarity"]**

Author's response to 1.4: it is now corrected in the revised version of the manuscript.

Response to Reviewer 2 are structured as follow: (1) 2.X: comments from Reviewer 2, (2) Response to 2.X: author's response and author's changes in manuscript when any. For sake of clarity, line and page from the first submission is used.

**Reviewer#2**

**This paper documents the improvements in land surface simulations driven by ERA-5 relative to ERA-Interim. The work is relevant from the standpoint of documenting the improvements in ERA-5 as it relates to hydrology. The manuscript is somewhat sloppily prepared (a figure was missing!). My comments and suggestions are below.**

We thanks anonymous Reviewer 2 for his/her review of the manuscript and for highlighting the relevance of the study for documenting the improvements in ERA-5 as it relates to hydrology. It is the main objective of the study.

We are sorry that a figure (figure 5 according to Reviewer#2 first major comment) was missing in the version of the manuscript he/she had. From the pdf file available on HESSD website, this figure is available, at least when downloaded from all co-author's institutes from the date Reviewer#2's comments were posted on-line (03/05/2018).

Reviewer#2 has made several fruitful comments/corrections/suggestions that led to an improved version of the manuscript. Again we would like to thanks Reviewer#2 for his/her work.

Responses to the Reviewer are available in the supplement.

**Major comments:**

**2.1. [The text has numerous language issues and grammar mistakes, some of which are listed below. I ran out of steam documenting all of them. I assume that the author would take a fresh and careful look at the manuscript to correct them all (including the ones that are not listed). Note that Figure 5 was missing in the manuscript version that I reviewed.]**

Author's response to 2.1
Many thanks for correcting some language issues and grammar mistakes, all the points listed by Reviewer#2 were corrected and a fresh and careful look at the revised version of the manuscript has been taken. We are sorry that figure 5 was missing in the version of the manuscript Reviewer#2 had. From the pdf file available on HESSD website, this figure is available, at least when downloaded from all co-author's institutes from the date Reviewer#2's comments were posted on-line (03/05/2018).

**2.2. [The paper is really an offline LDAS simulation, but makes no mention of other LDAS work. The literature review should encompass the recent work in this regard that have considered the assimilation of land measurements (NCA-LDAS, for.e.g). I think the paper would be more powerful and of broader interest if the authors can document how the ERA-5 forced system compares with the such LDAS efforts. For example, does ERA-5 have comparable skills to NLDAS2, the defacto standard land product over CONUS? How does it compare to MERRA2 and ERA-Interim-Land? Without such comparisons, the paper sounds more as a technical report the impact of ERA related changes.]**

Author's response to 2.2
This study only considers offline simulations and is not "[...] really an offline LDAS simulation [...]"; no assimilation of any land measurements is done in this work. The fact that only open loop

(offline) simulations are considered is what permits to express an evaluation on the quality of the used forcing being ERA-5 and ERA-Interim, while in case there would be an LDAS attached the causal attribution would become more complex. Furthermore, LDAS's notions appear only in the discussions and conclusions section L.510 as a next working step.

Although we agree that comparisons with other products are of interest, it goes beyond the scope of this study that is documenting the improvements in ERA-5 with respect to ERA-Interim as it relates to hydrology. It is likely that future work from the same group of authors will consider such comparisons especially with the shortcoming land version of ERA-5; ERA5-land (Sabater et al., 2018, workshop paper). We understand however the importance of mentioning other LDAS work and the following sentence has been modified:

P.16, L.350-352: "Albergel et al., 2017, 2018 (in prep.) recently presented a Land Data Assimilation System (LDAS-Monde) able to sequentially assimilate satellite derived estimates of surface soil moisture and LAI."
**is now:**
"ERA-5 has a great potential to further improve the representation of land surface variables if used to force offline LDAS. In the past recent years, several LDAS have emerged at different spatial scales, (i) regional like the Coupled Land Vegetation LDAS (CLVLDAS, Sawada and Koike, 2014, Sawada et al., 2015), the Famine Early Warning Systems Network (FEWSNET) LDAS (FLDAS, McNally et al., 2017), (ii) continental like the North American LDAS (NLDAS, Mitchell et al., 2004; Xia etal., 2012), the National Climate Assessment LDAS (NCA-LDAS Kumar et al., 2018) as well as at (iii) global scale like the Global Land Data assimilation (GLDAS, Rodell et al., 2004) and more recently LDAS-Monde (Albergel et al., 2017, 2018 in prep). LDAS-Monde is a global capacity system able to sequentially assimilate satellite derived estimates of surface soil moisture and LAI."

New references:
-Albergel, C., S. Munier, A. Bocher, C. Draper, D. J. Leroux, A. L. Barbu, J.-C. Calvet: LDAS-Monde global capacity integration of satellite derived observations applied over North America: assessment, limitations and perspectives. to be sumitted to Remote Sensing, Special Issue "Assimilation of Remote Sensing Data into Earth System Models", 2018

-Kumar, S.V., M. Jasinski, D. Mocko, M. Rodell, J. Borak, B. Li, H. Kato Beaudoing, and C.D. Peters-Lidard: NCA-LDAS land analysis: Development and performance of a multisensor, multivariate land data assimilation system for the National Climate Assessment. J. Hydrometeor., 0, https://doi.org/10.1175/JHM-D-17-0125.1

-McNally, A., Arsenault, K., Kumar, S., Shukla, S., Peterson, P., Wang, S., Funk, C., Peters-Lidard, C. D. and Verdin, J. P.: A land data assimilation system for sub-Saharan Africa food and water security applications. Scientific Data, 4, 170012, :10.1038/sdata.2017.12, 2017.

-Mitchell, K. E., et al. The multi-institution North American Land Data Assimilation System (NLDAS): Utilizing multiple GCIP products and partners in a continental distributed hydrological modeling system, J. Geophys. Res.,109, D07S90, 2004. doi:10.1029/2003JD003823

-Muñoz-Sabater, Joaquín, Emanuel Dutra, Gianpaolo Balsamo, Souhail Boussetta, Ervin Zsoter, Clement Albergel, Anna Agusti-Panareda: ERA5-Land: an improved version of the ERA5 reanalysis land component. Joint ISWG and LSA-SAF Workshop, 26-28 June 2018, Lisbon, Portugal.

-Rodell, M., P. R. Houser, U. Jambor, J. Gottschalck, K. Mitchell, C.-J. Meng, K. Arsenault, B. Cosgrove, J. Radakovich, M. Bosilovich, J. K. Entin, J. P. Walker, D. Lohmann, and D. Toll, The Global Land Data Assimilation System, Bull. Amer. Meteor. Soc., 85(3), 381–394, 2004.

-Sawada, Y., T. Koike, and J. P. Walker, A land data assimilation system for simultaneous simulation of soil moisture and vegetation dynamics, J. Geophys. Res.Atmos., 120, doi: 10.1002/2014JD022895, 2015.

-Sawada, Y., and T. Koike, Simultaneous estimation of both hydrological and ecological parameters in an ecohydrological model by assimilating microwave signal, J. Geophys. Res. Atmos., 119, doi:10.1002/2014JD021536, 2014.

-Xia, Y., et al. 2012, Continental-scale water and energy flux analysis and validation for the North American Land Data Assimilation System project phase 2 (NLDAS-2): 1. Intercomparison and application of model products, J. Geophys. Res., 117, D03109, doi:*10.1029/2011JD016048*, 2012.

**2.3. [I found the use of metrics to be a bit convoluted and inconsistent. For example, why use NIC for metrics such as R, instead of simply taking a difference? NIC is more useful when the dynamic range of the metrics are really large (NSE ranges from -infinity to 1, so a large negative value would blow up a domain average). I suggest sticking with simple differences so that the impact on the model runs are more intuitive. Why use ubRMSED to look at impacts on fluxes? (It is used for soil moisture because of the large climatological differences)Also, I would change the sign of N_MAE and N_ubRMSD to be the same as that of NIC values (Positive value indicating improvements).]**

Author's response to 2.3

Agreed, more consistency can now be found across the metrics and several changes have been made in the revised versions of the manuscript including:
- NIC is applied to NSE, only, and the impact on R is assessed using R differences,
- RMSD (instead of ubRMSD) is now used to assess the impact on fluxes (it is kept for soil moisture, for the reason explained by Reviewer#2, and snow evaluation as we believe it is still very informative since it shows the improvements on the "random" component of the error).
- Changing the sign of Nmae and NubRMSE is a bit contra-intuitive, instead we have decided to remove those metrics. Snow impact is now assesed using bias, ubRMSD and R.
- For each metrics, the 95% confidence interval of the median derived from a 10000 samples bootstrapping is provided.

It led to several changes in section 3.1 detailed below (text, tables and figures). A new Table (Table III) has been added to present scores for the snow evaluation, figure 6 (for snow evaluation) has been modified, figure 7 (panels c and d) now shows RMSD (instead of ubRMSD) and figure 11 (now figure 10) shows score differences (instead of N_RMSD and NIC_R). Please see also Author's response to 2.24.

The whole new section 3.1 is now (modification are highlighted in yellow):
"This section presents the results of the comparison versus in situ observations of land surface variables from model simulations using either ei_S, e5ei_S or e5_S starting with soil moisture. The statistical scores for 2010–16 surface soil moisture from ei_S, e5ei_S and e5_S are presented in Table II. Median R values on volumetric time-series (anomaly time series) along with their 95% confidence intervals are $0.66\pm0.02$ ($0.53\pm0.02$), $0.69\pm0.02$ ($0.54\pm0.04$) and $0.71\pm0.02$ ($0.58\pm0.03$) while median ubRMSD are $0.052\pm0.003$, $0.052\pm0.002$ and $0.050\pm0.003$ for ei_S, e5ei_S and and e5_S, respectively. These results underline the better capability of the ISBA LSM to represent surface soil moisture variability when forced by ERA-5 reanalysis. Also the latest configuration (e5_S) presents more stations with better R values on volumetric time-series (anomaly time series) than both ei_S and e5ei; respectively 60% and 75% (out of 110 and 107 stations, respectively). This is also reflected on figure 2 illustrating correlations values on volumetric time-series (fig.2a) and anomaly time-series (fig.2b) on maps. Stars symbols represent stations for which ISBA LSM performs best when forced ERA-Interim, circles when it is forced by ERA-5 with ERA-Interim precipitations and downward pointing triangles when it is forced by all ERA-5 atmospheric variables. Both maps on figure 2 are dominated by downward pointing triangles. Fig.2c(d) shows

histograms of R differences on volumetric (anomaly) time-series, for soil moisture from e5_S (in red) e5ei_S (in green) with respect to ei_S, median values of the differences are reported, also.

172 out of 344 gauging stations retained for the evaluation according to the criteria described in the methodology section presents NSE scores in the [-1, 1] interval. Figure 3 represents performance of each dataset for this pool of stations. Fig3.a is a scatterplot of NSE scores between in situ and simulated river discharges $Q$; NSE scores for $Q$ simulated with either ERA-5 but ERA-Interim precipitations (e5ei_S, green crosses) or ERA-5 (e5_S, red dots) function of NSE scores for $Q$ simulated using ERA-Interim (ei_S). When considering e5_S, almost all the red dots are above the 1:1 diagonal suggesting a general improvement from the use of e5_S. For a large part, e5ei_S green crosses are above this diagonal, suggesting that the improvement in e5_S does not only comes from precipitation but from other variables, also. Median NSE values are 0.06±0.06, 0.12±0.07 and 0.24±0.05 for ei_S, e5ei_S and e5_S, respectively. Fig.3b shows an histogram of river discharges ratio for ei_S (Qr_ei in blues), e5ei_S (Qr_e5ei in green) and e5_S (Qr_e5 in red), median values are 0.67, 075 and 0.77, respectively. While all three experiments underestimate $Q$ (a value of 1 being a perfect match), the use of e5ei_S and e5_S leads to better results. Finally, figure 3c illustrates hydrographs for a river station in Lousiana (33.08°N, -93.85°W) representing scaled $Q$ (using either observed or simulated drainage areas), in situ data (black crosses), simulated river discharges from ei_S (blue solid line), e5ei_S (green solid line) and e5_S (red solid line). From this hydrograph, the added value of e5_S is clear, particularly for the 2011 and 2015 main events. NSE scores are 0.47, 0.61 and 0.76 for ei_S, e5ei_S and e5_S, respectively. Figure 4 illustrates the added value of using e5_S (a) or e5ei_S (b) with respect to ei_S. For 156 out of the pool of 172 stations $NIC_{NSE}$ values computed using e5_S with respect to ei_S are positive (large blue circles) showing an general improvement from the use of e5_S (representing 91% of the stations) with a median $NIC_{NSE}$ value of 14%±0.05. When considering e5ei_S versus ei_S, they are still 118 (69%) with a median $NIC_{NSE}$ value of 4%±0.02 suggesting that the improvement in e5_S does not only comes from precipitation but from other variables, also. It is also worth-noticing that stations where a score degradation is observed (large red circles) are located in areas known for irrigation which is not represented in ISBA. All scores computed for seasons (December-January-Februray, March-April-May, Jun-July-August, September-October-November) suggest the same ranking (not shown).

The mean snow depth bias of ei_S (see Figure 5) highlights a clear underestimation of winter snow depth accumulation mainly over the Rocky Mountains. This is likely a result of the underestimation of snowfall by ei_S associated with an overestimation of snow melt due to the coarse resolution of the ei_S reflected in a smooth topography. The replacement of all forcing variables by e5_S but keeping ei_S precipitation (e5ei_S, Fig.5b) shows a slight increase in snow depth. This result justifies the above hypothesis that part of the snow underestimation is also due to temperature issues linked with a coarse model orography. Moving to the full e5_S forcing there is a clear increase of snow depth, when compared with both ei_S and e5ei_S forced simulations resulting from an increase in snowfall in e5_S. Figure 6 presents the mean seasonal cycle of bias and ubRMSD (fig.6a) and correlations (fig.6b) over 2010-2016. In addition to the added values of e5_S in terms of the mean snow depth already presented in figure 5, the temporal variability and random errors are also improved. Comparably with what was discussed for the mean bias, e5ei_S shows some benefits when compared with ei_S in terms of ubRMSD and correlation (median bias, ubRMSD and R values of e5_ei over the whole period are; -1.70±0.33 cm., 7.40±0.65 cm. and 0.77± 0.01, respectively, for ei_S they are; -2.11±0.33 cm., 7.58±0.65 cm. and 0.75± 0.01, respectively) while e5_S has a clear improvement in ubRMSD and correlation (median bias, ubRMSD and R values of e5_ei over the whole period are; -0.64±0.19 cm., 7.00±0.65 cm. and 0.82± 0.01, respectively). The improvements on the snow depth simulations are consistent throughout the entire snow covered season (see Fig.6a and b) with a maximum improvement from January to March. These results highlight the cumulative effect of the forcing quality on the snow depth simulation. Finally Table III presents scores from the comparison of snow depth with in situ measurements, median Bias, ubRMSD and R values are given for the three seasons affected by snow (September-October-November, December-January-February and Mars-April-May) and for the whole period. e5_S

always presents better scores when compared to ei_S and it is always the configuration presenting the highest percentage of stations with the best scores. Looking at the 95% confidence interval, for the correlation and bias it is clear that the changes are significant.
Results from the comparisons between ei_S, e5ei_S, e5_S and in situ sensible and latent flux measurements are presented in table IV and illustrated by figure 7 and 8. 37 stations present significant correlation values (at p-value < 0.05). For sensible heat flux, median correlation and RMSD values are 0.62±0.11, 0.62±0.11 and 0.65±0.11, 39.58±3.71 W.m$^{-2}$, 32.89±3.86 W.m$^{-2}$ and 32.73±2.61 W.m$^{-2}$ for ei_S, e5ei_S and e5_S, respectively. For latent heat flux, they are 0.63±0.05, 0.62±0.07 and 0.70±0.04, 39.00±5.38 W.m$^{-2}$, 37.12±4.37 W.m$^{-2}$ and 36.66±4.94 W.m$^{-2}$. As for surface soil moisture, river discharge and snow depth, e5_S presents better results than e5ei_S and ei_S. At the station level, figure 7 illustrates scatter plots of correlations and RMSD for sensible and latent heat flux from ei_S, e5ei_S, e5_S against in situ measurements of sensible (fig.7a for correlation, fig.7c for RMSD) and latent (fig.7b for correlation, fig.7d for RMSD) heat flux. Scores for either e5ei_S (green dots) or e5_S (in red) are presented function of those for ei_S. When looking at the correlations, almost all of e5_S and e5ei_S symbols (in red and green, respectively on fig.7a, fig.7c) are above the 1:1 diagonal indicating that e5_S and e5ei_S better represent sensible and latent heat flux than ei_S. Same tendency is observed for RMSD with most of the symbol below the 1:1 diagonal. If RMSD values are comparable for e5_S and e5ei_S, R values are clearly higher for e5_S. ''

**Table I : Evaluation datasets and associated metrics used in this study.**

| Datasets used for the evaluation | Source | Metrics associated |
|---|---|---|
| In situ measurements of soil moisture (USCRN, Bell et al., 2013) | https://www.ncdc.noaa.gov/crn | R (on both volumetric and anomaly time-series) ubRMSD |
| In situ measurements of streamflow (USGS) | https://nwis.waterdata.usgs.gov/nwis | Nash Efficiency (NSE), Normalized Information Contribution (NIC) based on NSE, Ratio of simulated and observed streamflow (Q) |
| In situ measurements of snow depth (GHCN, Menne et al., 2012a, b) | https://www.ncdc.noaa.gov/climate-monitoring/ | R, bias and ubRMSD |
| In situ measurements of sensible and latent heat fluxes (FLUXNET-2015) | http://fluxnet.fluxdata.org/data/fluxnet2015-dataset/ | R, RMSD |
| Satellite derived surface soil moisture (ESA CCI SSM v4, Dorigo et a., 2015, 2017) | http://www.esa-soilmoisture-cci.org | R (on both volumetric and anomaly time-series) |
| Satellite derived Leaf Area Index (GEOV1, Baret et al., 2013) | http://land.copernicus.eu/global/ | R and RMSD |
| Satellite-driven model estimates of land evapotranspiration (GLEAM, Martns et al., 2017) | http://www.gleam.eu | R and RMSD |
| Upscaled estimates of Gross Primary Production (GPP, Jung et al., 2017) | https://www.bgc-jenna.mpg.de/geodb/projects/Home.php | R and RMSD |

**Table II: Comparison of surface soil moisture with in situ observations for ei_S, e5ei_S and e5_S over 2010-2016 (April to September months are considered). Median correlations R (on volumetric and anomaly time series) and ubRMSD are given for the USCRN. Scores are given for significant correlations with p-values <0.05.**

|  | Median R* on volumetric time series, 95 % Confidence Interval** ( % of stations for which this configuration is the best) | Median R*** on anomalies time series, 95 % Confidence Interval** ( % of stations for which this configuration is the best) | Median ubRMSD* (m$^3$m$^{-3}$), 95 % Confidence Interval** ( % of stations for which this configuration is the best) |
|---|---|---|---|
| ei_S | 0.66±0.02 (20 %) | 0.53±0.02 (15 %) | 0.052±0.003 (19 %) |
| e5ei_S | 0.69±0.02 (20 %) | 0.54±0.04 (10 %) | 0.052±0.002 (24 %) |
| e5_S | 0.71±0.02 (60 %) | 0.58±0.03 (75 %) | 0.050±0.003 (57 %) |

**\* only for stations presenting significant R values on volumetric time series (p-value<0.05): 110 stations**
**\*\* 95% confidence interval of the median derived from a 10000 samples bootstrapping**
**\*\*\* only for stations presenting significant R values on anomaly time series (p-value<0.05): 107 stations**

**Table III: Comparison of snow depth with in situ measurements, median Bias, ubRMSD and R values are given for the three seasons affected by snow (SON, DJF, MAM) and for the whole period (All). SON, DJF and MAM stand for September-October-November, December-January-February and Mars-April-May, respectively.**

| | | Median bias (cm)*, 95 % Confidence Interval** ( % of stations for which this configuration is the best) | Median ubRMSD (cm)*, 95 % Confidence Interval** ( % of stations for which this configuration is the best) | Median R*, 95 % Confidence Interval** ( % of stations for which this configuration is the best) |
|---|---|---|---|---|
| ei_S | SON | -0.27±0.04 (13 %) | 2.05±0.17 (13 %) | 0.70±0.01 (21 %) |
| | DJF | -6.28±0.86 (11 %) | 10.34±0.63 (17 %) | 0.72± 0.01 (20 %) |
| | MAM | -1.90±0.33 (15 %) | 7.82±0.79 (17 %) | 0.65± 0.01 (18 %) |
| | All | -2.11±0.33 (11 %) | 7.58±0.65 (14 %) | 0.75± 0.01 (19 %) |
| e5ei_S | SON | -0.25±0.04 (12 %) | 2.03±0.15 (10 %) | 0.74± 0.01 (23 %) |
| | DJF | -4.84±0.80 (14 %) | 9.98±0.50 (14 %) | 0.75± 0.01 (21 %) |
| | MAM | -1.49±0.33 (14 %) | 7.61±0.76 (13 %) | 0.69±0.02 (22 %) |
| | All | -1.70±0.33 (14 %) | 7.40±0.65 (14 %) | 0.77± 0.01 (20 %) |
| e5_S | SON | -014±0.03(76 %) | 1.83±0.14 (77 %) | 0.79± 0.01 (56 %) |
| | DJF | -1.70±0.44 (75 %) | 9.64±0.46 (69 %) | 0.80± 0.01 (59 %) |
| | MAM | -0.57±0.22 (71 %) | 7.43±0.79 (70 %) | 0.76± 0.01 (60 %) |
| | All | -0.64±0.19 (75 %) | 7.00±0.65 (72 %) | 0.82± 0.01 (61 %) |

**\* only for stations presenting more than 80% of (daily) data; 1901 out of 2056 stations.**
**\*\* 95% confidence interval of the median derived from a 10000 samples bootstrapping**

**Table IV: Comparison of sensible (H) and latent (LE) heat flux with in situ observations for ei_S, e5ei_S and e5_S. Median correlations (R) and median RMSD are given for the fluxnet stations. Scores are given for significant correlations with p-values <0.05.**

|  | H Median R*, 95 % Confidence Interval** ( % of stations for which this configuration is the best) | H Median RMSD* W.m⁻², 95 % Confidence Interval** ( % of stations for which this configuration is the best) | LE Median R*, 95 % Confidence Interval** ( % of stations for which this configuration is the best) | LE Median RMSD* W.m⁻², 95 % Confidence Interval** ( % of stations for which this configuration is the best) |
|---|---|---|---|---|
| ei_S | 0.62±0.11 (8 %) | 39.58±3.71 (5 %) | 0.63±0.05 (8 %) | 39.00±5.38 (16 %) |
| e5ei_S | 0.62±0.11(27 %) | 32.89±3.86 (27%) | 0.62±0.07 (11 %) | 37.12±4.37 (22 %) |
| e5_S | 0.65±0.11 (65 %) | 32.73±2.61 (68 %) | 0.70±0.04 (81 %) | 36.66±4.94 (62 %) |

\* only for stations presenting significant R values (p-value<0.05): 37 stations
\*\* 95% confidence interval of the median derived from a 10000 samples bootstrapping

[Figure]

**Figure 6: (a) Mean seasonal cycle of the bias (dashed lines) and ubRMSD (solid lines) averaged over all stations and (b) the mean seasonal cycle of the correlations for ei_S (in blue), e5ei_S (in green) and e5_S (in red).**

[Figure]

**Figure 7: Scatterplots illustrating evaluation of ei_S, e5ei_S, e5_S against in situ measurements of sensible (a for correlation, c for RMSD) and latent (b for correlation, d for RMSD) heat flux. Scores for either e5ei_S (green dots) or e5_S (in red) are presented as function of those for ei_S.**

[Figure]

**Figure 10: RMSD differences (a, c, e) and Correlation differences (b, d, f) for e5_S simulations with respect to ei_S simulations for three land surface variables: evapotranspiration, Gross Primary Production and Leaf Area Index from top to bottom. Areas in red represent an improvement from the use of ERA-5.**

**2.4. [Section 3.1: I think the descriptions need to tone down the language on how much improvements are actually gained. From table 1, it looks like the improvements are quite small though they are systematic with the new version. I think it is important to quantify the magnitude of improvements (showing their spatial distribution through, for e.g., histograms).]**

Author's response to 2.4
In agreement with Reviewer#2's comment, some parts of section 3.1 are now re-worded in the revised version of the manuscript (please see also Author's response to 2.3, 2.24). The idea that improvements, even when they are quite small are systematic is now mentioned in the conclusion, also.

We believe that several figures already quantify the magnitude of improvements (e.g. figure 3.a, 4, 5, 6 and 7). Two histograms of R differences, as suggested by Reviewer#2, where added to figure 2

(panels c & d) to show the spatial distribution of the improvement on correlation for soil moisture (for both volumetric and anomaly time series). Please see below new figure 2.

P.12, L.388: the following sentence has been added to describe the new figure: " Fig.2c(d) shows histograms of R differences on volumetric (anomaly) time-series, for soil moisture from e5_S (in red) e5ei_S (in green) with respect to ei_S, median values of the differences are reported, also."

[Figure]

**Figure 2 : Maps of correlation (R) on volumetric time-series (a) and anomaly time-series (b) between in situ measurements at 5 cm depth from the USCRN network and the ISBA Land Surface Model within the SURFEX modeling platform forced by either ERA-Interim (ei_S), ERA-5 with ERA-Interim precipitations (e5ei_S) and ERA-5 (e5_S). For each stations presenting significant R (p-values < 0.05) simulation that presents the better R values is represented. Stars symbols are when ei_S, presents the best value, circles when it e5ei_S and downward pointing triangles when it is e5_S. (c) Shows histograms of R differences on volumetric time-series, R(e5_S)-R(ei_S) in red and R(e5ei_S)-R(ei_S) in green, median values of the differences are reported, also. (d) Same as (c) for R values on anomaly time-series.**

**Minor comments:**

**2.5. [Fix the sentence starting with 'ERA-5 important changes ..' to something like ' ERA-5 has important changes relative to ERA-Interim former atmospheric reanalysis including...']**

Author's response to 2.5
It has now been fixed in the revised version of the manuscript

**2.6. [Change the sentence 'ERA-5 is forseen .. ' to something like 'As ERA-5 is expected to replace ERA-Interim reanalysis, this study assesses whether']**

Author's response to 2.6

P.2, L.65-69 "It will eventually replace ERA-Interim reanalysis. Assessing ERA-5 ability to force a LSM with respect to ERA-Interim is therefore highly relevant. To that end, ERA-5, ERA-Interim as well as a combination of both (ERA-5 with precipitation of ERA-Interim) are used to constrain [...]"

is now:

"As ERA-5 will eventually replace ERA-Interim reanalysis assessing its ability to force a LSM with respect to ERA-Interim is highly relevant. In this study, ERA-5, ERA-Interim as well as a combination of both (ERA-5 with precipitation of ERA-Interim) are used to constrain [...]"

**2.7. [Change the sentence 'ERA-5 impact on the ISBA ..' to 'ERA-5's impact on ISBA LSM relative to ERA-Interim is evaluated using remote sensing...' ]**

Author's response to 2.7

P.1, L.22-24: "ERA-5 impact on the ISBA LSM with respect to ERA-Interim is assessed over a data-rich area: North America. A comprehensive evaluation of ERA-5 impact is conducted using remote sensing and in-situ observations covering a substantial part of the land surface storage and fluxes."

is now :

"ERA-5 impact on ISBA LSM relative to ERA-Interim is evaluated using remote sensing and in-situ observations covering a substantial part of the land surface storage and fluxes over the CONtinuous US (CONUS) domain."

**2.8. [Line 34 – Fix 'Interim ..' to 'Interim.' (only one period).]**

Author's response to 2.8 : done

**2.9. [Line 36: change 'extend' to 'extent' ]**

Author's response to 2.9 : done

**2.10. [Line 46: Change 'essentials' to 'essential']**

Author's response to 2.10 : done

**2.11. [Line 52: Change 'progresses' to 'progress']**

Author's response to 2.11 : done

**2.12. [Line 55: Add a comma after 'decade'.]**

Author's response to 2.12 : done

**2.13. [Lines 58-60: MERRA is retired. More appropriate to refer to MERRA2 papers. Given that this paper focuses on land-only simulations, there should be a description of LDAS analysis forced by observed precipitation (and meteorology) such as NLDAS, GLDAS ,etc.]**

Author's response to 2.12

Following changes have been made in the revised version of the manuscript:

"Amongst them are NASA's Modern Era Retrospective-analysis for Research and Applications (MERRA; Rienecker et al., 2011) as well as ECMWF's (European Centre for Medium-Range Weather Forecasts) Interim reanalysis (ERA-Interim; Dee et al., 2011). Their offline use in LSMs

led to global Land Surface Variables (LSVs) reanalysis datasets that can support e.g. water resources analysis (Schellekens et al., 2017), like MERRA-Land (Reichle, 2011) and ERA-Interim/Land (Balsamo et al., 2015)."

is now:

"Amongst them are NASA's Modern Era Retrospective-analysis for Research and Applications (MERRA; Rienecker et al., 2011 and MERRA2; Gelaro et al. 2016,) as well as ECMWF's (European Centre for Medium-Range Weather Forecasts) Interim reanalysis (ERA-Interim; Dee et al., 2011). Their offline use in either LSMs or Land Data Assimilation System (LDAS), with or without meteorological corrections (e.g., precipitations) led to global land surface variables (LSVs) reanalysis datasets that can support e.g. water resources analysis (Schellekens et al., 2017), like MERRA-Land and MERRA2-Land (Reichle, 2011; 2017), ERA-Interim/Land (Balsamo et al., 2015), the forthcoming ERA5-Land (Muñoz-Sabater et al., 2018), the North American LDAS (NLDAS, Mitchel et al., 2004), the Global LDAS (GLDAS, Rodell et al., 2004) and LDAS-Monde (Albergel et al., 2017)."

Author's response to 2.14 :
P.2, L.65-68, "ERA-5 important changes relative to ERA-Interim former atmospheric reanalysis include a higher spatial and temporal resolution as well as a better global balance of precipitation and evaporation."
is now :
"ERA-5 has important changes relative to ERA-Interim former atmospheric reanalysis including a higher spatial and temporal resolution as well as a better global balance of precipitation and evaporation."

**2.15. [Lines 96: Change to say 'Section 2 presents the details of two atmospheric ..']**

Author's response to 2.15 : done

**2.16. [Line 120: Change to say 'which allows it to use ..']**

Author's response to 2.16 : done

**2.17. [Line 132: Add a comma after 'study']**

Author's response to 2.17 :
Sentence is now : "This study makes use of the $CO_2$-responsive version of the ISBA LSM included in the open-access SURFEX modelling platform of Météo-France (Masson et al., 2013)."

**2.18. [Section 2.3: I would say 'interpolated to' rather than 'interpolated at': What interpolation methods were used?]**

Author's response to 2.18 :
A bi-linear interpolation from the native reanalysis grid to the regular grid, it is now added in the revised version of the manuscript (L.202).

**2.19. [Line 217: Kumar et al. (2009) is not in the list of references.]**

Author's response to 2.19
Reference to Kumar et al. (2009) is now in the list of references along with Kumar et al. (2018) that we find appropriate in this context.

- Kumar, S.V, R. H. Reichle, R. D. Koster, W. T. Crow, and C. Peters-Lidard. 2009. "Role of Subsurface Physics in the Assimilation of Surface Soil Moisture Observations." *J. Hydrometeor*, 10 (6): 1534-1547 [10.1175/2009JHM1134.1]
- Kumar, S.V., M. Jasinski, D. Mocko, M. Rodell, J. Borak, B. Li, H. Kato Beaudoing, and C.D. Peters-Lidard: NCA-LDAS land analysis: Development and performance of a multisensor, multivariate land data assimilation system for the National Climate Assessment. J. Hydrometeor., 0, https://doi.org/10.1175/JHM-D-17-0125.1

**2.20. [Line 237: I would not say 'artificially increasing the perceived agreement' – Just that the skill values are higher because it includes the seasonal cycle. ]**

Author's response to 2.20 : done
P.7, L.236-239: "Soil moisture time series usually show a strong seasonal pattern possibly artificially increasing the perceived agreement between modeled and observed data sets." is now : "Soil moisture time series usually show a strong seasonal pattern possibly increasing the skill values between modeled and observed data sets."

**2.21. [Line 239: 'Monthly averaged are also computed' (?)]**

Author's response to 2.21 : Please see Author's answer to 2.23 which provides clarification on this paragraph.

**2.22. [Line 240: Change 'week' to 'weeks'.]**

Author's response to 2.22 : done

**2.23. [Lines 239-242: It sounds like this you are really computing the z-scores rather than anomalies, since you are scaling the differences with standard deviation]**

Author's response to 2.23 :

According to Reviewer#2's comments 2.21 and 2.23, this paragraph has been clarified as follow:

P.7/8, L236-239: "To avoid seasonal effects, time series of anomalies from a moving monthly averaged are also computed. At each grid and observation points, the difference to the mean is calculated using a sliding window of five week and the difference is scaled by the standard deviation as in Albergel et al., (2013b). Anomaly time series reflect the time-integrated impact of antecedent meteorological forcing."
is now:
"To avoid seasonal effects, monthly anomaly time-series are calculated. At each grid and observation point, the difference from the mean is produced for a sliding window of five weeks, and the difference is scaled to the standard deviation as in Albergel et al., (2013b). For each surface soil moisture estimate at day (i), a period F is defined, with F= [i−17,i+17](corresponding to a five-week window). If at least five measurements are available in this period, the average soil moisture value and the standard deviation are calculated. Anomaly time series reflect the time-integrated impact of antecedent meteorological forcing."

**2.24. [Line 245: What significance test is done to compute the p-values? This varies depending on the metric of interest. In particular, since several derived metrics (NICs) are used here, how did you compute the statistical significance?]**

Author's response to 2.24 :
The p-values is applied on correlation values and only stations with significant correlation values (at p-values < 0.05) are retained, it is now clarified in the revised version of the manuscript. Table II on soil moisture evaluation now shows the 95% confidence interval for all metrics (95% confidence interval of the median derived from a 10000 samples bootstrapping). Please see Author's response to 2.3, also.

**2.25. [Line 265: Change from 'an NSE' to 'a NSE']**

Author's response to 2.25 : done

**2.26. [Line 323: Change 'exercises' to 'studies']**

Author's response to 2.26 : done

**2.27. [Line 347: Change 'equivalents' to 'equivalent']**

Author's response to 2.27 : done

**2.28. [Figure 1: As the authors describe, this figure is not very useful. The lines are too close to each other in most part. It will be easier to see them if you plot the differences (relative to ei_S ; then you only have two lines). Another option is to show a seasonal cycle rather than the entire time series.]**

Author's response to 2.28
Agreed, seasonal cycles would prove better, a new figure 1 along with a new caption (please see below) has been produced. Text has also been slightly modified to match with the new figure.

P.12, L.364-365: "Averaged time-series of the six main land surface variables evaluated in this study over the whole domain for 2010-2016 are illustrated on figure 1, [...]"
is now:

"Seasonal time-series of the six main land surface variables evaluated in this study over the whole domain for 2010-2016 are illustrated on figure 1, [...]"

[Figure]

*Illustration 1: Seasonal cycle for the 6 main land surface variables evaluated in this study over the whole domain for 2010-2016: (a) river discharge, (b) snow depth, (c) leaf area index , (d) liquid soil moisture in the second layer of soil (1-4 cm depth), (e) evapotranspiration and (f) gross primary production. Land surface variables simulated with SURFEX forced by ERA-Interim (ei_S) are in blue, by ERA-5 (e5_S) with precipitation from ERA-Interim (e5ei_S) in blue and by ERA-5 in red.*

**2.29. [Lines 390-395: Say NSE rather than 'efficiency']**

Author's response to 2.29 : done

**2.30. [Lines 465: Change 'Aprils' to 'April']**

Author's response to 2.30 : done

**2.31. [Lines 479-480: What does 'lasting dataset' mean?]**

Author's response to 2.30 :
"the three lasting dataset" has now been removed from the revised version of the manuscript.

P.15, L.476-480, "Figure 10 illustrates seasonal scores between ISBA LSM forced by either ERA-Interim (ei_S  in blue) ERA-5 but ERA-Interim precipitation (e5ei in green) or ERA-5 (e5_S  in red) ***and the three lasting dataset***; (fig10.a, fig10.b) evapotranspiration estimates from the GLEAM

project over 2010-2016, (fig10.c, fig10.d) upscaled GPP from the FLUXCOM project over 2010-2013 and (fig10.e, fig10.f) LAI estimates from the Copernicus GLS project over 2010-2016. Left column (fig10.a, c and e) are for RMSD and right column (fig8.b, d, e) for correlations."
is now :
"Figure 9 illustrates seasonal scores between ISBA LSM forced by either ERA-Interim (ei_S in blue) ERA-5 but ERA-Interim precipitation (e5ei in green) or ERA-5 (e5_S in red) for; (fig9.a, fig9.b) evapotranspiration estimates from the GLEAM project over 2010-2016, (fig9.c, fig9.d) upscaled GPP from the FLUXCOM project over 2010-2013 and (fig9.e, fig9.f) LAI estimates from the Copernicus GLS project over 2010-2016. Left column (fig9.a, c and e) are for RMSDs and right column (fig9.b, d, e) for correlations."

**2.32. [Lines 509-510: Fix – 'It is however acknowledge that ..']**

Author's answer to 2.32: done

[revised manuscript text omitted]

60 Amongst them are NASA's Modern Era Retrospective-analysis for Research and Applications (MERRA; Rienecker et al., 2011 and MERRA2; Gelaro et al. 2016,) as well as

65 ECMWF's (European Centre for Medium-Range Weather Forecasts) Interim reanalysis (ERA-Interim; Dee et al., 2011). Their offline use in either LSMs or Land Data Assimilation System (LDAS), with or without meteorological corrections (e.g., precipitations) led to global Land Surface Variables (LSVs) reanalysis datasets that can support e.g. water resources analysis (Schellekens et

al., 2017), like MERRA-Land and MERRA2-Land (Reichle, 2011; 2017), ERA-Interim/Land (Balsamo et al., 2015), the forthcoming ERA5-Land (Muñoz-Sabater et al., 2018), the North American LDAS (NLDAS, Mitchel et al., 2004), the Global LDAS (GLDAS, Rodell et al., 2004) and LDAS-Monde (Albergel 
[revised manuscript text omitted]
). A bi-linear interpolation from the native reanalysis grid to the regular grid has been used. For all three experiments, the first year (2010) was spun up 20 times to allow the model to reach equilibrium. Comparing e5_S to ei_S provides the overall improvements from ERA-Interim to ERA-5. The idealized e5ei_S simulation was carried out to assess the role of precipitation changes from ERA-Interim to ERA-5.

220    This study makes use of several in-situ measurement data sets as well as satellite-derived estimates of Earth Observations that are described in the next two sections. The different performance metrics used for the evaluation are described, also. Their choice is of crucial interest; it is governed by the nature of the variable itself and is influenced by the purpose of the investigation and its sensitivity to the considered variables (Stanski et al. 1989). No single metric or statistic can capture all the
225    attributes of environmental variables. Some are robust in respect to some attributes while insensitive to others (Entekhabi et al. 2010). While performance metrics like the correlation coefficient, unbiased root mean squared differences, root mean squared differences, efficiency score (depending on the considered variable) are first applied to the three simulations independently, metrics like the Normalized Information Contribution (NIC, e.g. Kumar et al., 2009) are then used to quantify
230    improvement or degradation from a data set to another. Table I summarises the different dataset used for the evaluation as well as the performance metrics used.

**2.3.1. *In situ measurement of soil moisture, river discharges, snow depth and fluxes**

USCRN is a network of climate-monitoring stations maintained and operated by the National Oceanic and Atmospheric Administration (NOAA). It aims at providing climate-science-quality
235    measurements of air temperature and precipitation. To increase the network's capability of monitoring soil processes and drought, soil observations were added to USCRN instrumentation. In 2011, the USCRN team completed at each USCRN station in the conterminous United States the installation of triplicate-configuration soil moisture and soil temperature probes at five standards

depths (5, 10, 20, 50, and 100 cm) as prescribed by the World Meteorological Organization. 111
240  stations present data between 2009 and 2016. Stations provide data at an hourly time step. Similar
to prior study, datasets potentially affected by frozen conditions were masked out using an observed
temperature threshold of 4ºC (e.g. Albergel et al., 2013a). The second layer of soil of ISBA between
1 and 4 cm depth (the diffusion scheme is used in this study) is compared to in situ measurements at
5 cm depth at a three hourly time step (model output) between April and September in order to
245  avoid as much as possible frozen conditions. The ability of ei_S, e5ei_S and e5_S to reproduce
surface soil moisture variability is first assessed using the correlation coefficient (R) and unbiased
Root Mean Square Differences (ubRMSD). Climatology differences between model and in-situ
observations make a direct comparison difficult (Koster et al., 2009b).

250   Soil moisture time series usually show a strong seasonal
pattern possibly increasing the skill values between modeled and observed data sets.

255   To
avoid seasonal effects, monthly anomaly time-series are calculated. At each grid and observation
point, the difference from the mean is produced for a sliding window of five weeks, and the
difference is scaled to the standard deviation as in Albergel et al., (2013b). For each surface soil
moisture estimate at day (i), a period F is defined, with F= [i−17,i+17](corresponding to a five-
260  week window). If at least five measurements are available in this period, the average soil moisture
value and the standard deviation are calculated. Anomaly time series reflect the time-integrated
impact of antecedent meteorological forcing. The latter is mainly reflected in the upper layer of soil.
The correlation coefficient is also computed for anomaly time-series ($R_{ano}$). For correlations, the p-
value (a measure of the correlation significance) is also calculated indicating the significance of the
265  test (as in Albergel et al. 2010), and only cases where the p-value is below 0.05 (i.e., the correlation
is not a coincidence) are retained. Stations with non-significant R values can be considered suspect
and are excluded from the computation of the network average metrics. This process may remove
some reliable stations too (e.g., in areas where the model might not realistically represent soil
moisture).
270  Over 2010-2016 river discharge from ei_S, e5ei_S and e5_S are compared to daily streamflow data
from the U.S. Geological Survey (USGS; http://nwis.waterdata.usgs.gov/nwis). Data are chosen for
sub-basins with large drainage areas (10,000km$^2$ or greater) and with a long observation time series

(4 years or more). Smaller basins are excluded due to the low resolution of CTRIP (0.5°x0.5°). It is common to express observed and simulated river discharge (Q) data in m³s⁻¹. Given that the observed drainage areas may differ slightly from the simulated ones, specific discharge in mm.d⁻¹ (the ratio of Q to the drainage area) is used in this study, similarly to Albergel et al., 2017. Stations with drainage areas differing by more than 20% from the simulated ones are also discarded. This criterion aims to ensure a meaningful comparison between observed and simulated values. It is necessary for coping with the significant distortions in the model representation of the river network that are caused by the coarse spatial resolution of the CTRIP global river network (0.5°x0.5°). Impact on Q is evaluated using the efficiency score (NSE) (Nash and Sutcliff, 1970). NSE evaluates the model ability to represent the monthly discharge dynamics and is given by:

$$NSE = 1 - \frac{\sum_{t=1}^{T} \left(Q_s^t - Q_o^t\right)^2}{\sum_{t=1}^{T} \left(Q_o^t - \overline{Q_o^t}\right)^2} \tag{1}$$

where $Q_s^t$ is the simulated river discharge (by either ei_S, e5ei_S or e5_S) at time t and $Q_o^t$ is observed river discharge at time t, T is the total number of days and $\overline{Q_o^t}$ is the average observed discharge. NSE can vary between $-\infty$ and 1. A value of 1 corresponds to identical model predictions and observed data. A value of 0 implies that the model predictions have the same accuracy as the the mean of the observed data. Negative values indicate that the observed mean is a more accurate predictor than the model simulation. Only stations with a NSE greater than -1 for the benchmark experiment, ei_S, are considered, leading to 172 stations over the considered domain. A Normalized Information Contribution (NIC as in Kumar et al. ; 2009) measure is then computed to quantify the improvement or degradation due to the specific atmospheric reanalysis used to force ISBA. The $NIC_{NSE}$ values are computed for both e5_S and e5ei_S with respect to ei_S as:

$$NIC_{NSE(e5;5ei)} = \frac{NSE_{(e5;e5ei)} - NSE_{(ei)}}{1 - NSE_{(ei)}} \tag{2}$$

The NIC_{NSE} metric provides a normalized measure of the improvement through the use of either NSE_{e5ei} or NSE_{e5} as a fraction of the maximum possible skill improvement (1-NSE_{ei}).  Positive and negative NIC_{NSE} values indicate improvements and degradations in either e5_S or e5ei_S relative to ei_S river discharges estimates, respectively. NICs along with their 95% confidence interval of the median derived from a 10000 samples bootstrapping are provided for e5_S, e5ei_S. The ratio of simulated and observed river

discharges is computed also $\left(Q_s^t/Q_o^t\right)$, the closer to one it is, the better the simulated river discharges are.

The Global Historical Climatology Network (GHCN) Daily dataset, developed to meet the needs of climate analysis and monitoring studies that require data at a daily time resolution contains records from over 75000 stations in 179 countries and territories (Menne et al., 2012a, b). Numerous daily variables are provided, including maximum and minimum temperature, total daily precipitation, snowfall, and snow depth. In this study, over North America, stations with daily snow depth data from 2010-2016, with less than 10% missing and at least 15 days of snow presences per year on average (to avoid using stations always reporting zero snow depth) are used, it results in 1901 stations out of 20. The ability of ei_S, e5ei_S and e5_S to reproduce snow depth and its variability is assessed using the bias, correlation coefficient (R) and unbiased Root Mean Square Difference (ubRMSD)

Daily observations of sensible and latent heat fluxes from the FLUXNET-2015 dataset with at least 2-yr of data are used over 2010-2016 to evaluate e5_S, e5ei_S and ei_S ability to reproduce flux variability. The FLUXNET-2015 dataset includes data collected at sites from multiple regional flux networks as well as several improvements to the data quality control protocols and the data processing pipeline (http://fluxnet.fluxdata.org/data/fluxnet2015-dataset/). 37 stations are retained for the evaluations, two metrics are considered: R and RMSD.

Performance metrics are applied to each individual station of each network; thereafter, network metrics are computed by providing the median values of the statistics from the individual stations within each network. For each metrics, the 95% confidence interval of the median derived from a 10000 samples bootstrapping is provided.

[revised manuscript text omitted]

 ERA-5 has a great potential to further improve the representation of land surface variables if used to

570  force offline LDAS. In the past recent years, several LDAS have emerged at different spatial scales, (i) regional like the Coupled Land Vegetation LDAS (CLVLDAS, Sawada and Koike, 2014, Sawada et al., 2015), the Famine Early Warning Systems Network (FEWSNET) LDAS (FLDAS, McNally et al., 2017), (ii) continental like the North American LDAS (NLDAS, Mitchell et al., 2004; Xia etal., 2012), the National Climate Assessment LDAS (NCA-LDAS Kumar et al., 2018) as

575  well as at (iii) global scale like the Global Land Data assimilation (GLDAS, Rodell et al., 2004) and more recently LDAS-Monde (Albergel et al., 2017, 2018 in prep). LDAS-Monde is a global capacity system able to sequentially assimilate satellite derived estimates of surface soil moisture and LAI. TheyAlbergel et al. (2017) found that the main improvements of their analysis (i.e. with assimilation) when compared to an open-loop experiment (simple model run) were linked to

580  vegetation variables and the assimilation of vegetation estimates. They have also proposed further advances on a better use of satellite-based microwave data in the assimilation system. Having LDAS-Monde analysis forced by ERA-5 atmospheric forcing should both combined the strengths of an improved atmospheric reanalysis on the terrestrial water cycle and of the assimilation of satellite derived products on the vegetation cycle. Effort will now be concentrate on the use of

585  ERA-5 and strengthening LDAS-Monde through the direct assimilation of satellite-based soil moisture and vegetation properties from microwave remote sensing. It will enable fostering links with potential applications like climate reanalysis of the LSVs as well as going from a monitoring system of the LSVs and extreme events (like agricultural drought) to a forecasting system. Preliminary results suggest that a LSVs forecast initialized by an analysis is more robust than one

590  initialized by a simple model run (Albergel et al., 2018, in prep). Preliminary tests over Europe also indicate similar benefits from the use of ERA-5 (not shown). When the whole ERA-5 period will be available (1979-present), in addition with the availability of the ERA-5 10-member Ensemble of Data Assimilation (at lower spatial and temporal resolution though), it will be possible to develop a global long term ensemble of land surface variables reanalysis forced by high quality atmospheric

595  data. It will make it possible providing uncertainties in the representation of the atmospheric forcing, while land surface variables may require special considerations and perturbation methods. Capturing those uncertainties coming from the simplifications and assumptions in the LSM is of paramount interest for many applications from monitoring to forecasting.

**Acknowledgments-***Results where Generated using Copernicus Climate Change Service*

600  *Information 2017. E. Dutra work was supported by the Portuguese Science Foundation (FCT) under project IF/00817/2015.*

[revised manuscript text omitted]

* only for stations presenting significant R values on volumetric time series (p-value<0.05): 110 stations
** 95% confidence interval of the median derived from a 10000 samples bootstrapping
*** only for stations presenting significant R values on anomaly time series (p-value<0.05): 107 stations

**Table III: Comparison of snow depth with in situ measurements, median Bias, ubRMSD and R values are given for the three seasons affected by snow (SON, DJF, MAM) and for the whole period (All). SON, DJF and MAM stand for September-October-November, December-January-February and Mars-April-May, respectively.**

| | | Median bias (cm)*, 95 % Confidence Interval** ( % of stations for which this configuration is the best) | Median ubRMSD (cm)*, 95 % Confidence Interval** ( % of stations for which this configuration is the best) | Median R*, 95 % Confidence Interval** ( % of stations for which this configuration is the best) |
|---|---|---|---|---|
| ei_S | SON | -0.27±0.04 (13 %) | 2.05±0.17 (13 %) | 0.70±0.01 (21 %) |
| | DJF | -6.28±0.86 (11 %) | 10.34±0.63 (17 %) | 0.72± 0.01 (20 %) |
| | MAM | -1.90±0.33 (15 %) | 7.82±0.79 (17 %) | 0.65± 0.01 (18 %) |
| | All | -2.11±0.33 (11 %) | 7.58±0.65 (14 %) | 0.75± 0.01 (19 %) |
| e5ei_S | SON | -0.25±0.04 (12 %) | 2.03±0.15 (10 %) | 0.74± 0.01 (23 %) |
| | DJF | -4.84±0.80 (14 %) | 9.98±0.50 (14 %) | 0.75± 0.01 (21 %) |
| | MAM | -1.49±0.33 (14 %) | 7.61±0.76 (13 %) | 0.69±0.02 (22 %) |
| | All | -1.70±0.33 (14 %) | 7.40±0.65 (14 %) | 0.77± 0.01 (20 %) |
| e5_S | SON | -014±0.03(76 %) | 1.83±0.14 (77 %) | 0.79± 0.01 (56 %) |
| | DJF | -1.70±0.44 (75 %) | 9.64±0.46 (69 %) | 0.80± 0.01 (59 %) |
| | MAM | -0.57±0.22 (71 %) | 7.43±0.79 (70 %) | 0.76± 0.01 (60 %) |
| | All | -0.64±0.19 (75 %) | 7.00±0.65 (72 %) | 0.82± 0.01 (61 %) |

* only for stations presenting more than 80% of (daily) data; 1901 out of 2056 stations.
** 95% confidence interval of the median derived from a 10000 samples bootstrapping

**Table IH: Comparison of sensible (H) and latent (LE) heat flux with in situ observations for ei_S, e5ei_S and e5_S. Median correlations (R) and median RMSD are given for the fluxnet stations. Scores are given for significant correlations with p-values <0.05.**

920

| | H Median R*, 95 % Confidence Interval** ( % of stations for which this configuration is the best) | H Median RMSD* W.m⁻², 95 % Confidence Interval** ( % of stations for which this configuration is the best) | LE Median R*, 95 % Confidence Interval** ( % of stations for which this configuration is the best) | LE Median RMSD* W.m⁻², 95 % Confidence Interval** ( % of stations for which this configuration is the best) |
|---|---|---|---|---|
| ei_S | 0.62±0.11 (8 %) | 39.58±3.71 (5 %) | 0.63±0.05 (8 %) | 39.00±5.38 (16 %) |
| e5ei_S | 0.62±0.11(27 %) | 32.89±3.86 (27%) | 0.62±0.07 (11 %) | 37.12±4.37 (22 %) |
| e5_S | 0.65±0.11 (65 %) | 32.73±2.61 (68 %) | 0.70±0.04 (81 %) | 36.66±4.94 (62 %) |

**\* only for stations presenting significant R values (p-value<0.05): 37 stations**
**\*\* 95% confidence interval of the median derived from a 10000 samples bootstrapping**

[Figure]

**Illustration 1: Seasonal** cycle **of the 6 main land surface variables evaluated in this study over the whole domain for 2010-2016 :: (a) river discharge, (b) snow depth, (c) leaf area index , (d) liquid soil moisture in the second layer of soil (1-4 cm depth), (e) evapotranspiration and (f) gross primary production. Land surface variables simulated with SURFEX forced by ERA-Interim (ei_S) are in blue, by ERA-5 (e5_S) with precipitation from ERA-Interim (e5ei_S) in blue and by ERA-5 in red.**

925

[Figure]

**Figure 2 : Maps of correlation (R) on volumetric time-series (a) and anomaly time-series (b) between in situ measurements at 5 cm depth from the USCRN network and the ISBA Land Surface Model within the SURFEX modeling platform forced by either ERA-Interim (ei_S), ERA-5 with ERA-Interim precipitations (e5ei_S) and ERA-5 (e5_S). For each stations presenting significant R (p-values < 0.05) simulation that presents the better R values is represented. Stars symbols are when ei_S, presents the best value, circles when it e5ei_S and downward pointing triangles when it is e5_S. (c) Shows histograms of R differences on volumetric time-series, R(e5_S)-R(ei_S) in red and R(e5ei_S)-R(ei_S) in green, median values of the differences are reported, also. (d) Same as (c) for R values on anomaly time-series.**

930

935

[Figure]

**Figure 3: (a) Scatterplots of efficiency scores between in situ and simulated river discharges $Q$; efficiency scores for $Q$ simulated with SURFEX forced either by ERA-5 but ERA-Interim precipitations (e5ei_S, green crosses) or ERA-5 (e5_S, red dots) function of efficiency scores for $Q$ simulated using ERA-Interim (ei_S). (b) Histograms of river discharges ratio for ei_S (Qr_ei in blues), e5ei_S (Qr_e5ei in green) and e5_S (Qr_e5 in red). (c) Hydrograph for a river station in Lousiana (33.08°N, 1.52°W) representing scaled $Q$ (using either observed or simulated drainage areas), in situ data (black crosses), simulated river discharges from ei_S (blue solid line), e5ei_S (green solid line) and e5_S (red solid line).**

[Figure]

**Figure 4: Normalized Information Contribution scores based on efficiency scores (NIC_NSE) (a) e5_S with respect to ei_S and (b) e5ei_S with respect to ei_S. Small dots represent station for which the benchmark experiment (ei_S) present efficiency scores smaller than -1, large circles when it presents values higher than -1. Positive values (blue large circles) suggest an improvement over ei_S, negative values (red large circles) a degradation. For sack of clarity, a factor 100 has been applied on NIC.**

[Figure]

**Figure 5: Mean snow depth bias for December-January-February in ei_S (a) and differences between e5ei_S and ei_S (b), e5_S and e5ei_S (c), e5_S and ei_5 (d).**

[Figure]

**Figure 6:**  (b)  ows the sean seasonal cycle of the bias (dashed lines) and ubRMSD (solid lines) averaged over all stations and (b) the mean seasonal cycle of the correlations for ei_S (in blue), e5ei_S (in green) and e5_S (in red).

[Figure]

**Figure 7: Ss**catterplots illustrating evaluation of ei_S, e5ei_S, e5_S against in situ measurements  sensible (a for correlation, c for RMSD) and latent (b for correlation, d for RMSD) heat flux. Scores for either e5ei_S (green dots) or e5_S (in red) are presented  function of those for ei_S.

990

~~Figure 8: Normalized Information Contribution scores based on correlations values between in situ measurements from the fluxnet sites data and (a) e5_S with respect to ei_S for sensible heat flux and (b) e5_S with respect to ei_S for latent heat flux. Normalised ubRMSD values between in situ measurements from the fluxnet sites data and (c) e5_S with respect to ei_S for sensible heat flux and (d) e5_S with respect to ei_S for latent heat flux. Blue circles indicate improvement compared to ei_S (positive values of NIC_R in a,b, and negative values of N_ubRMSD in c,d) whereas red circles correspond to a degradation (negative values of NIC_R in a,b, and positive values of N_ubRMSD in c,d). For sack of clarity a factor 100 has been applied on NIC_R.~~

995

1000

[Figure]

**Figure 89 : Seasonal correlations on (a) volumetric time-series and (b) anomaly time-series between surface soil moisture estimates from the ESA CCI project (ESA-CCI SSM v4) and soil moisture from the second layer of soil of the ISBA LSM forced by ERA-Interim (ei_S in blue), ERA-5 but with precipitation from ERA-Interim (e5ei_S in green) and ERA-5 (e5_S in red) over 2010-2016. Maps of correlations differences between soil moisture from e5_S and ei_S on volumetric time-series (c) and anomaly time-series (d), areas in red represent an improvement from the use of ERA-5. Grey areas represents areas that where flagged out for elevation greater than 1500 m above sea level.**

[Figure]

 **Figure 910: Seasonal scores between ISBA LSM within SURFEX forced by either ERA-Interim (ei_S in blue) ERA-5 but ERA-Interim precipitation (e5ei_S in green) or ERA-5 (e5_S in red) and (a, b) evapotranspiration estimates from the GLEAM project over 2010-2016, (c, d) upscaled GPP from the FLUXCOM project over 2010-2013 and (e, f) LAI estimates from the Copernicus GLS project over 2010-2016. Left column (a, c and e) are for RMSD and right column (b, d, e) for correlations.**

[Figure]

**Figure 1:**  **RMSD** differences **(a, c, e) and** Correlation differences **(** **b, d, f)**  **for e5_S simulations with respect to ei_S simulations for three land surface variables**: **evapotranspiration, Gross Primary Production and Leaf Area Index from top to bottom. Areas in red represent an improvement from the use of ERA-5.** R